# A Survey on Application of Non-Orthogonal Multiple Access to Different Wireless Networks

**Asim Anwar** [1,*,†] , **Boon-Chong Seet** [2,†] , **Muhammad Amish Hasan** [3,†] **and Xue Jun Li** [2,†]

1    Department of Technology, The University of Lahore, Lahore 54590, Pakistan
2    Department of Electrical and Electronic Engineering, Auckland University of Technology, Auckland 1010, New Zealand; boon-chong.seet@aut.ac.nz (B.-C.S.); xuejun.li@aut.ac.nz (X.J.L.)
3    Department of Electronics and Electrical Systems, The University of Lahore, Lahore 54590, Pakistan; amish.hassan@es.uol.edu.pk
*    Correspondence: asim.anwar@tech.uol.edu.pk
†    These authors contributed equally to this work.

**Abstract:** The fifth generation (5G) wireless systems are anticipated to meet unprecedented capacity and latency requirements. In order to resolve these challenges in 5G, non-orthogonal multiple access (NOMA) is considered as a promising technique due to its ability to enhance spectrum efficiency and user access. As opposed to conventional orthogonal multiple access (OMA) which relies on orthogonal resource sharing, NOMA has a potential of supporting a higher number of users by multiplexing different users in the same resource in a non-orthogonal manner. With advanced receiver techniques, such as successive interference cancellation (SIC), the intra-user interference can be minimized at the NOMA receiver. To date, there are comprehensive surveys on NOMA, which describe the integration of NOMA with different communication technologies and discuss different NOMA classifications. However, the existing literature is scarce in reviewing state-of-the-art applications of NOMA from the perspective of its application to cellular networks (CNs), device-to-device (D2D) communications, and wireless sensor networks (WSNs). Therefore, the purpose of this survey is to fill this gap in knowledge. Specifically, NOMA with its underlying concepts are elaborated in detail. In addition, detailed system model of different NOMA-based wireless networks is presented. Furthermore, irrespective of the underlying spatial topology of the considered NOMA-based wireless network, general analytical expressions are presented to characterize the network performance. Finally, some challenges related to NOMA design are highlighted and potential research directions are pointed out to address these issues.

**Keywords:** 5G; non-orthogonal multiple access; cellular networks; device-to-device communication; wireless sensor networks; outage probability; sum-rate maximization; interference

---

## 1. Introduction

The blistering growth in wireless devices, such as smart phones and tablets, coupled with rapid evolution in mobile Internet technology, have massively transformed the use of traditional analog voice telephony to resource ravenous multimedia applications. Subsequently, it is anticipated that there would be a thousand-fold increase in mobile data traffic by the year 2020 [1,2]. This explosive growth in mobile devices has ignited the demand for ubiquitous and seamless connectivity. Driven by these requirements, the wireless communication technology has been entering into an era of hyper-connected society through the advent of upcoming fifth generation (5G) systems and services. Particularly, the key stringent requirements of 5G systems which advocate its proposal can be stated as follows [3–5]: (1) To support demands of massive and ubiquitous connectivity by providing at least $10^6$ connections/Km$^2$,

which is ten times superior than the connectivity density of current fourth generation (4G) systems; (2) to meet the ultra low latency requirements of about $\leq 1$ ms; and (3) to enhance the spectral efficiency by 10–15 times over current 4G systems.

The last two decades have witnessed a paradigm shift in the development and use of wireless systems and technology. From an implementation perspective, the evolution of various wireless communications systems spans around using different multiple access schemes. More precisely, frequency division multiple access (FDMA), time division multiple access (TDMA), code division multiple access (CDMA), and orthogonal frequency division multiple access (OFDMA) have been utilized as the underlying multiple access (MA) schemes in first generation (1G), second generation (2G), third generation (3G), and fourth generation (4G) systems, respectively. From the working principle viewpoint, the aforementioned MA schemes belong to a class of orthogonal multiple access (OMA) [6]. An OMA can be formally defined as a MA scheme in which radio resources (frequency/time/code/their combination) are orthogonally allocated to different users. In particular, multiple users access the channel in an orthogonal fashion i.e., the users are orthogonal in frequency/time/code domain. As such, the direct consequence of using OMA is that there would be no interference among users' message signals. As a result, the key benefit of using OMA is that low-complexity and cost efficient-receivers are utilized to facilely decode the users' message signals. However, the prime limitation of adopting OMA is that it is severely limited in supporting a large number of users or connections. Consequently, this is would result in spectral inefficiency and is particularly not feasible to support massive connectivity and Internet of Things (IoT), which have been deemed as key features in 5G systems. Further, the random nature of wireless channel induce impairments which perpetually destroy the orthogonality among users in OMA systems [7]. As a result, high-complexity receivers need to be invoked in order to rejuvenate the orthogonality among users.

Despite the fact that 4G systems are rapidly growing and are in wide use, due to the scarcity of available spectrum resources, the 4G technology and services are still inept to meet the stringent requirements of the upcoming 5G systems [8–10]. In order to meet the rigorous demands of 5G systems, the ingenious proposal of non-orthogonal multiple access (NOMA) has been admitted as a latest member of MA family. The key idea of NOMA is that multiple users simultaneously share the same radio resource i.e., the users access the channel in a non-orthogonal manner. This would enhance the spectrum efficiency at the expense of increased receiver complexity, which is ineluctably required to separate users' messages. The superior spectral efficiency of NOMA over OMA techniques can be illustrated with the aid of following example. Consider a multiuser communication system in which there is a user with poor channel conditions that needs to send/receive a data with high priority. In order to maintain fairness among users, the conventional OMA based system would allocate an exiguous dedicated orthogonal resource to this user. Consequently, the spectral efficiency and overall system throughput would degrade. In contrast, NOMA would solve this problem by allowing multiple users to access the available radio resource simultaneously. This results in efficient spectrum utilization while concurrently maintaining the fairness among users [11]. Apart from improving spectral efficiency and fairness over conventional OMA, NOMA has a strong potential to improve the cell-edge throughput, requires relaxed channel feedback, and to significantly reduce the overall latency [12]. These properties make NOMA a suitable and promising MA candidate to support massive connectivity requirement in 5G systems [13,14].

The beauty and brilliance of the NOMA concept lies in a fact that it has excellent compatibility with other communication technologies. This allows NOMA to be readily integrated with the existing and future wireless systems. For instance, NOMA has proved to be concordant with the conventional OMA techniques, such as TDMA and OFDMA [15]. As a consequence, this remarkable feature offered by NOMA along with its potential to meet the demands of 5G systems has captured the attention from both academia and industry. Particularly, NOMA has been adopted as an underlying MA scheme by various standardization bodies. For example, third generation partnership project (3GPP) has developed a standard long-term evolution advanced (LTE-A) based on NOMA, where

NOMA is termed as a multiuser superposition transmission (MUST). More precisely, adopting NOMA allows multiple users to simultaneously share the same subcarrier in OFDMA without modifying the LTE resource blocks [16]. In addition, the latest Advanced Television Systems Committee (ATSC) standard for digital television (TV) broadcasting is based on the NOMA principle, where multiple data streams are superimposed in a same resource. Consequently, this would result in an enhanced spectral efficiency for the TV broadcasting system [17]. Moreover, apart from evaluating and analyzing basic principle of NOMA and proving its superiority over OMA theoretically, industries like Huawei Technologies and Nippon Telegraph and Telephone (NTT) DOCOMO have performed field trials and prototype evaluations to verify the analytical performance gains of NOMA over OMA. Interesting readers are referred to see the references [18–24]. The aforementioned advantages of NOMA and its appreciation from standardization bodies inarguably imply that NOMA possesses an immense potential to meet the requirements of 5G and beyond 5G systems.

To date, the existing state-of-the-art survey papers on NOMA can be broadly divided into two categories. The first category provides review of prior arts on NOMA from the perspective of different NOMA classifications such as power and code domain NOMA. The second category of survey papers review those works on NOMA which investigate the integration of NOMA with different communication technologies such as multiple-input multiple-output (MIMO), cooperative communication, millimeter waves, full duplex (FD), and so on [12,14,25–43]. However, to the best of our knowledge, there is no current survey which attempts to review state-of-the-art on NOMA according to its application in cellular networks (CNs), device-to-device (D2D) communication, and wireless sensor networks (WSNs). Therefore, the goal of this survey is to exhaustively review the prior arts on NOMA from the perspective of its application to CNs, D2D communication, and WSNs. In particular, apart from reviewing the state-of-the-art on NOMA-based CN, D2D communications, and WSNs, the main contributions of this survey are summarized as follows:

- A comprehensive network model is provided for each of the considered NOMA-based CN, D2D communication, and WSN.
- In order to evaluate the performance of each considered NOMA-based network, outage probability expressions are presented.
- In addition, the sum-rate maximization problem is formulated for each type of NOMA-based network, which could serve as a good reference point for beginners and practitioners working on NOMA-based CN, D2D communication, or WSN.

The rest of this paper is organized as follows. Section 2 provides the basic concepts of NOMA. The application of NOMA to CN, D2D communication, and WSN has been comprehensively reviewed in Section 3. The discussion of NOMA related research challenges and future directions is presented in Section 4. Finally, Section 5 concludes the survey. For quick reference, the organization of this survey is presented in Figure 1.

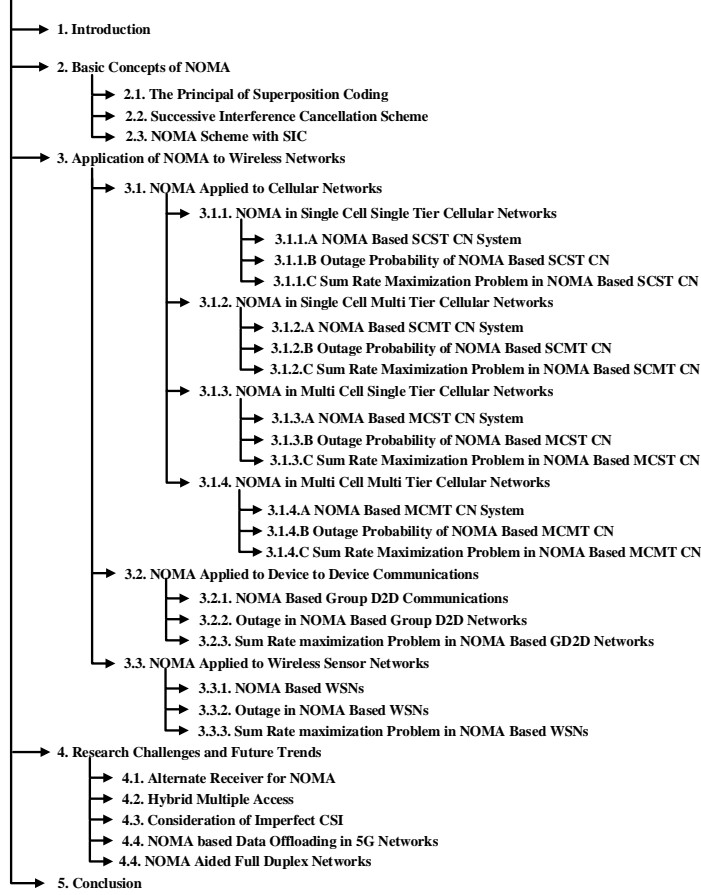

**Figure 1.** Organization of this survey.

## 2. Basic Concepts of NOMA

From conceptual and implementation perspectives, NOMA solutions can be classified into two broad categories, namely *code domain* and *power domain* NOMA [25]. The fundamental working principle of code domain NOMA is that multiple users are allowed to share the same resources (time/frequency), which is very similar to the operation of classic CDMA systems. However, code domain NOMA has maintained a vital difference with CDMA systems because in contrast to CDMA, it utilizes sparse spreading or non-orthogonal low cross-correlation sequences to realize multiplexing. Some popular code domain NOMA solutions include sparse code multiple access (SCMA), low-density spreading CDMA (LDS-CDMA), LDS aided OFDM (LDS-OFDM), MUST, and successive interference cancellation aided MA (SAMA). The readers are referred to see [44–55] and references therein to have more deep insights into the code domain NOMA.

The other classification of NOMA is based on power domain multiplexing. In contrast to code domain NOMA, the multiple users are superimposed in a same resource (time/frequency/code) by allocating different power levels to multiple users. Therefore, multiple users access the channel in a non-orthogonal fashion by applying the principle of superposition coding (SC). At the receiver side, advance multiuser detection (MUD) techniques such as successive interference cancellation (SIC) or dirty paper coding (DPC) are applied to decode the users' message signals [56,57]. Figure 2 presents the classification of NOMA schemes. In this paper, the main focus is to provide a comprehensive review for the application of power domain NOMA to CNs, D2D communications, and WSNs. Therefore, unless otherwise stated, NOMA refers to power domain NOMA in the remainder of this paper. In addition, SC and SIC techniques are the vital components of a NOMA system. Therefore, the rest of this section discusses the principle of SC, the working of SIC technique, and NOMA system with SIC receiver.

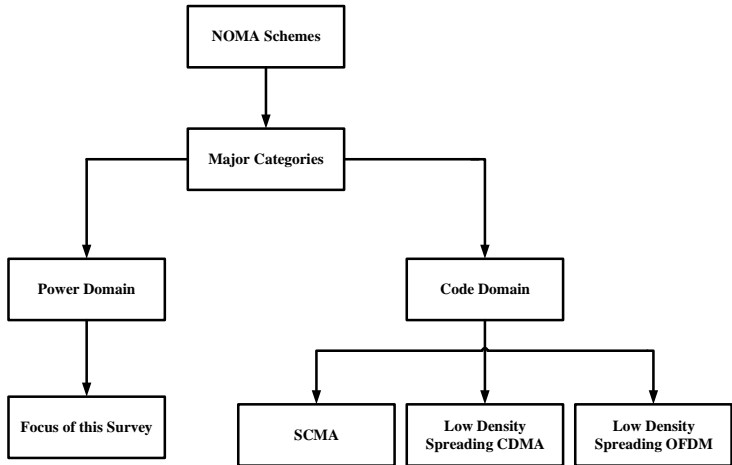

**Figure 2.** Classification of non-orthogonal multiple access (NOMA) schemes.

## 2.1. The Principle of Superposition Coding

The kind of communication scenario in which a single source needs to communicate with many receivers arises in various wireless systems, such as base station (BS) communicating with cellular users, WiFi access point sending information to multiple users, hotspot at airport or in shopping mall, military commander giving specific instructions to different units, and so on. The traditional technique to realize communication in these scenarios is to apply conventional OMA, where orthogonal channels are allocated to each user by utilizing time/frequency/code domain. From an information-thoretic perspective, this approach is generally not optimal in achieving the sum-rate or capacity region of the Gaussian broadcast channel (BC). As such, the non-orthogonal technique, known as SC, is considered as a promising scheme that has a potential for achieving the capacity of the Gaussian BC. In addition, SC is capable of enhancing the spectral efficiency of the overall system [58–60].

The idea of SC was first proposed and investigated in the pioneer work of [61], where SC is suggested as a method of simultaneously transmitting information from a single source to multiple receivers. The authors in [62] proposed and discussed different strategies for implementing SC. In addition, with the aid of experimental results, the authors also present the set of rate-pairs which are capable of achieving superior spectral efficiency compared to the OMA schemes. Some common examples of SC include broadcasting TV transmission, a speaker/teacher delivering a lecture to people with different aptitudes, and radio transmission. Specifically, SC is considered as a physical layer method to simultaneously transmit independent messages of multiple users. From the operational and implementation perspectives, SC technique can be regarded as a multi-layer modulation scheme in which each layer corresponds to the modulated and coded message signal of different users. The transmitter then superimposes independent users' message signals by adding them in order to constitute a composite signal before transmitting to the channel [63]. Figure 3 illustrates the basic transmitter implementing SC and the concept of SC based transmission system.

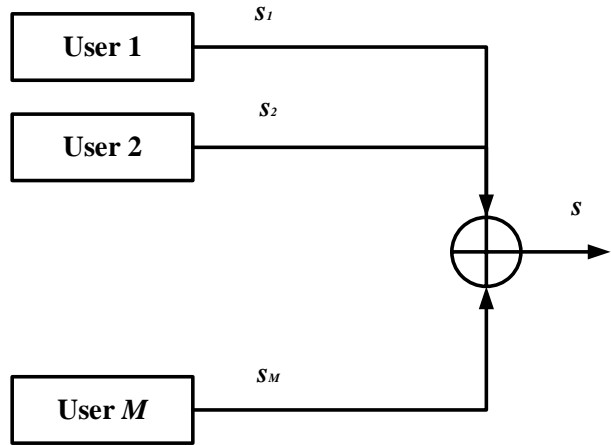

**Figure 3.** Transmitter implementing superposition coding (SC).

In order to obtain deep insights for understanding the principle of SC, more rigorous treatment is required. Consider a multiuser communication system with a single source and a total of $M$ users. Further, assume that the total transmission power of source is constrained by $P$ Watts and the message signal of user $m$, $m = 1, 2, ..., M$ is denoted by $s_m$. Note that depending on the type of communication system, $s_m$ could be a baseband message signal or obtained as a result of using some particular modulation and coding scheme. However, for notational simplicity and ease of exposition, the discussion herein considers $s_m$ as a message signal regardless of its underlying construction method. By applying the SC principle, a source generates a superimposed composite signal, denoted as $s$, which is then simultaneously transmitted to all $M$ users. The superimposed signal $s$ at the source can be written as:

$$s = \sum_{m=1}^{M} \sqrt{a_m P} s_m, \tag{1}$$

where $a_m$ is the power allocation coefficient of user $m$ and $\sum_{m=1}^{M} a_m = 1$. By inspecting Equation (1), it can be observed that SC introduces intra-user interference, which needs to be removed by each user $m$ before decoding its own message. As such, an efficient MUD technique is required to suppress the intra-user interference in order to fully enjoy the benefits of SC. Therefore, the next subsection discusses the working principle of SIC, which is considered as a promising and relatively low-complexity MUD scheme to be used at the users' receiver side [64].

*2.2. Successive Interference Cancellation Scheme*

In order to decode the user $m$ message from the superimposed signal $s$, the SIC scheme is regarded as a rudimental approach [65]. The basic structure of SIC is presented in Figure 4. The key operating principle of SIC relies on exploiting the power differences among users participating in SC [66]. The working of SIC can be explained as follows. First, the users are ordered in descending order of their signal strengths. As such, consider that the users' power allocation coefficients in Equation (1) are ordered as $a_1 \geq ... \geq a_m... \geq a_M$ i.e., users $m = 1$ and $m = M$ are allocated with maximum and minimum powers, respectively. As a result, the users from $1, ..., m - 1$ are regarded as *strong (higher order) users* while the users from $m + 1, ..., M$ are considered as *weak (lower order) users* for user $m$. Next, the decoding is performed in a successive fashion, as depicted in Figure 4. This essentially implies that user $m$ first successively decodes the message signals of all the higher order users starting from user 1 to $m - 1$. After decoding the higher order users, the user $m$ decodes its own message by treating the lower order users' messages as noise. Considering the perfect SIC operation (ideal cancellation

of higher order users), the signal-to-interference-noise-ratio (SINR) at user $m$, denoted as $\Gamma_m$ can be expressed as:

$$\Gamma_m = \frac{P_m}{\sum_{i=m+1}^{M} P_i + \sigma^2}, \tag{2}$$

where $P_i$ is the signal strength or power (typically received power) of user $i$, and $\sigma^2$ is the noise power. Note that the summation term in Equation (2) is actually the intra-user interference from users $m + 1, ..., M$ resulted due the application of SC. However, this intra-user interference is seen as a noise by user $m$ because its allocated power level $a_m P$ is larger than the power levels of all the weak users [67]. This assumption simplifies the decoding operation and implementation of SIC receiver but essentially increases the signal-to-noise-ratio (SNR) level required for successful decoding. With the brief introduction to SC and SIC, we are now in a position to discuss the concept of NOMA in next sub-section.

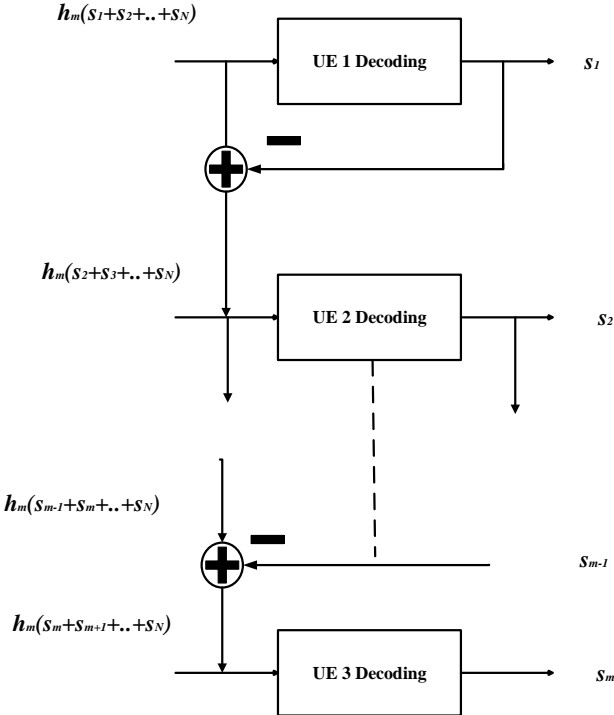

**Figure 4.** Block diagram of successive interference cancellation (SIC) receiver.

*2.3. NOMA Scheme with SIC*

Consider a single source which is simultaneously transmitting messages to total of $M$ NOMA users. The users are assumed to be randomly distributed inside the coverage of source. Further, denote by $h_m = \frac{g_m}{1 + d_m^\alpha}$ as the channel gain (power) between user $m$ and source. Here, the variables $g_m$ and $d_m$ represent the fading (power) gain and distance between the source and user $m$, respectively, and $\alpha$ is the pathloss exponent.

According to the NOMA principle, it allocates more power to users with poor channel conditions and vice-versa [68]. As such, in order to apply NOMA technique, the users are first required to be ordered at the source. Without loss of generality, consider that the users are sorted in ascending order of their channel gains as $h_1 \leq ... \leq h_M$. Consequently, the power allocation coefficients are sorted as $a_1 \geq ... \geq a_M$. The source then applies the principle of SC to generate a superimposed signal $s$ as given

by Equation (1), which is then sent to all the users simultaneously. At receiver of user $m$, the received signal, denoted by $r_m$, can be expressed as [69]:

$$r_m = \sum_{i=1}^{M} \sqrt{h_m a_i P} s_i + n_m, \tag{3}$$

where $n_m$ represents the additive white Gaussian noise (AWGN) at user $m$ receiver.

At the receiver of user $m$, the decoding is performed in ascending order of the users' channel gains. As a result, SIC is applied to decode the message signals of all the higher order users $1, ..., m-1$. After removing the interference from higher order users, the user $m$ decodes its own message by considering all the lower order users from $m+1, ..., M$ as noise. Consequently, based on Equation (2), the SINR at user $m$ can be expressed as:

$$\Gamma_m = \frac{h_m a_m P}{h_m P \sum_{i=m+1}^{M} a_i + \sigma^2} \tag{4}$$

$$= \frac{h_m a_m \rho}{h_m \rho \sum_{i=m+1}^{M} a_i + 1} \tag{5}$$

where $\rho = \frac{P}{\sigma^2}$ is the transmit SNR.

## 3. Application of NOMA to Wireless Networks

The previous section has provided an insight into the NOMA and its underlying basic concepts. With this background, the focus of this section is to present the application of NOMA to different types of wireless networks. More precisely, in the light of current literature, we will review NOMA applied to CN, D2D communication, and WSN. Our approach is to provide a basic structure and working of the considered NOMA-based wireless network in the context of existing literature, and then discuss useful analytical results which are important to evaluate the performance of NOMA-based wireless network. More precisely, we first review NOMA applied to the considered wireless network in the context of state-of-the-art prior works. Second, the wireless network model based on NOMA is discussed. Third, the performance of NOMA system under considered wireless network will be presented in terms of outage probability. Finally, a sum-rate maximization problem for NOMA will also be formulated.

### 3.1. NOMA Applied to Cellular Networks

From conceptual and operational viewpoints, a cellular network can be classified in the following four categories. (1) Single-cell single-tier (SCST), (2) single-cell multi-tier (SCMT), (3) multi-cell single tier (MCST), and (4) multi-cell multi-tier (MCMT) CNs. As such, the subsequent discussion comprehensively reviews the application of NOMA to the aforementioned classifications of CNs. In addition, for illustration purposes, Figures 5–8 present different types of NOMA-based cellular networks.

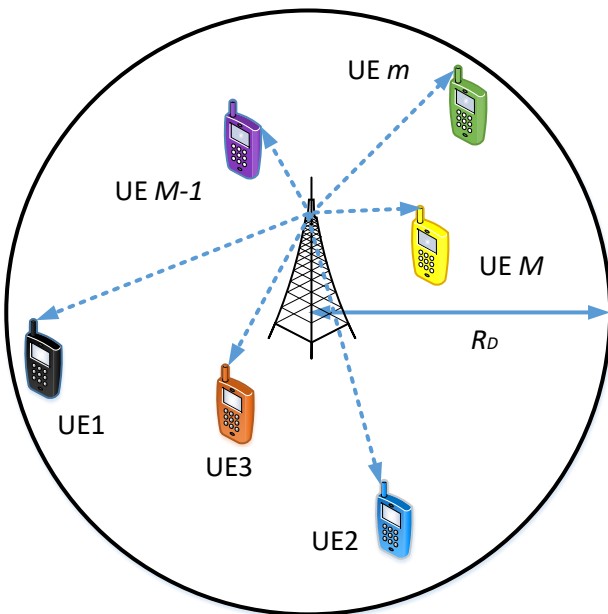

**Figure 5.** NOMA-based single-cell single-tier (SCST) cellular network.

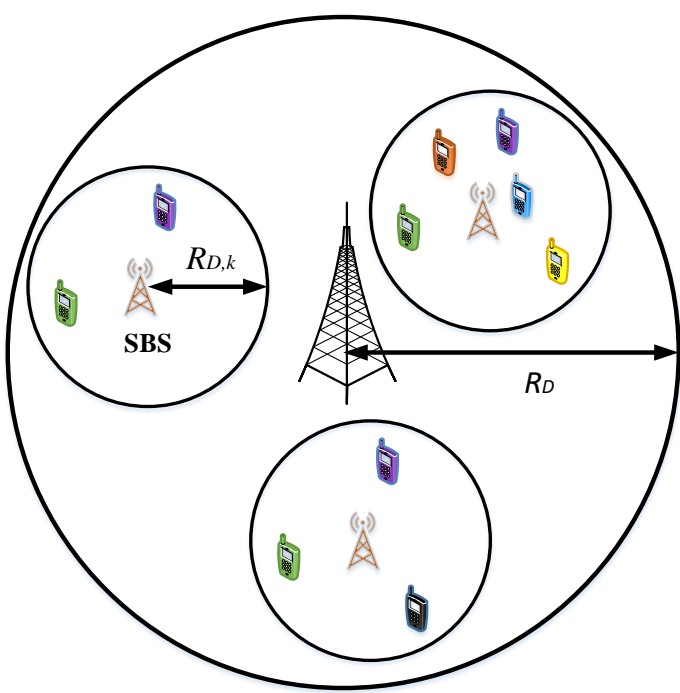

**Figure 6.** NOMA-based single-cell multi-tier (SCMT) cellular network.

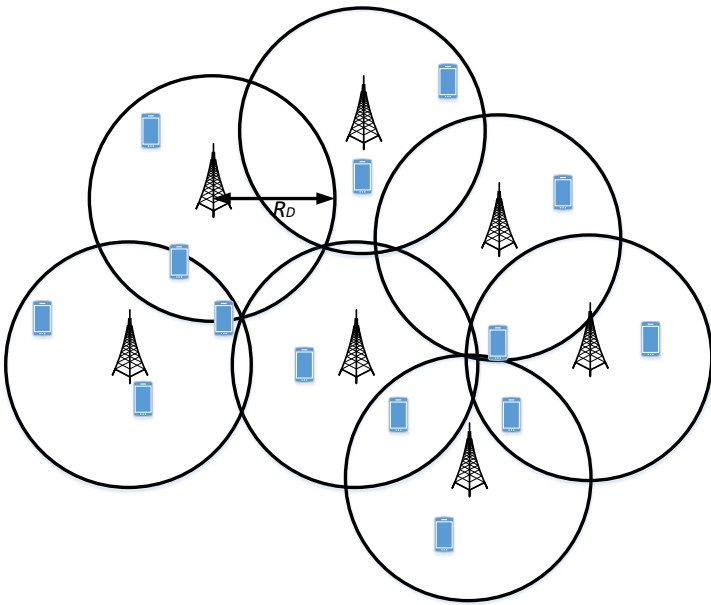

**Figure 7.** NOMA-based multi-cell single tier (MCST) cellular network.

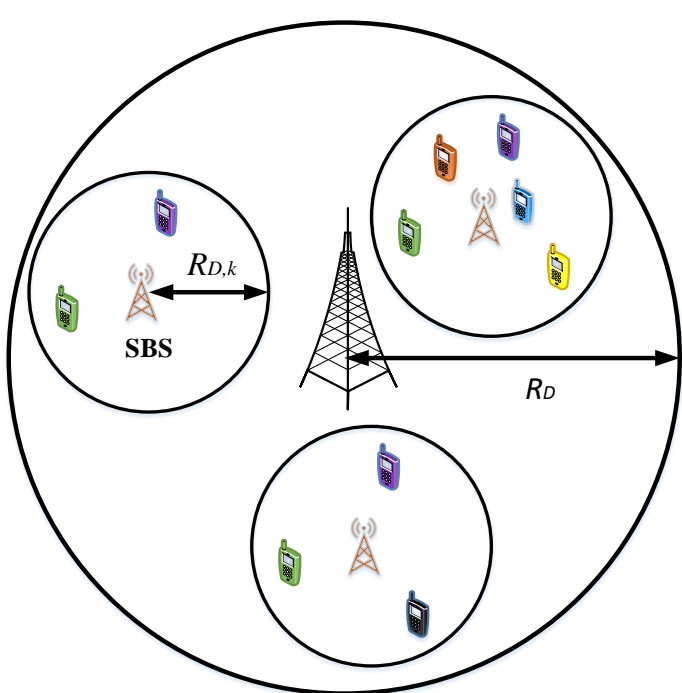

**Figure 8.** NOMA-based multi-cell multi-tier (MCMT) cellular network.

### 3.1.1. NOMA in Single-Cell Single-Tier Cellular Networks

The most common type of NOMA-based CNs which are exhaustively investigated in current literature are SCST CNs. The initial works of [20,21,56] proposed the concept of the NOMA technique with SIC receiver and validated their proposal by performing system level simulations. The authors reported robust performance gain of 30%–35% achieved by NOMA over OMA in terms of overall cell-throughput. In addition, the work in [21] also discussed the practical implementation issues related to the realization of NOMA such as power allocation, error propagation in SIC, signaling

overhead, and user mobility. The classical work of [70] could be regarded as a pioneer paper which investigated the performance of single-cell downlink NOMA system with randomly deployed users. Closed-form expressions for outage probability and ergodic sum-rate are derived to characterize the performance of the considered NOMA network. The authors in [20,56] integrated the LTE specific practical considerations with NOMA related operations. More precisely, the problem of user scheduling was resolved using proportional fairness (PF) scheduler. In addition, the critical issue of users' power allocation is addressed by applying dynamic transmit power allocation (TPA), fractional transmit power allocation (FTPA), and full-search power allocation (FSPA) techniques. Furthermore, hybrid automatic repeat request (HARQ) protocol was adopted for retransmissions when decoding error occurs.

The impact of user grouping on NOMA performance is investigated in [11,56,71–83]. The authors in [56] grouped users together whose channel gains differ significantly. This would facilitate users' power allocation using TPA/FPA, resulting in superior user and cell throughput over conventional OMA. In [11], the authors investigated the impact of user pairing by considering fixed power allocation NOMA (F-NOMA) and cognitive radio inspired NOMA (CR-NOMA). The authors derived closed-form analytical expressions for achievable sum-rate in order to evaluate the performance of the proposed user pairing on the considered NOMA systems. The results demonstrate that after careful user grouping (pairing users with significant difference in channel qualities), F-NOMA outperforms conventional OMA. This implies that when grouped users have similar channel conditions, there would be limited gain of NOMA over OMA. On the other hand, in CR-NOMA scheme, the users with low and high channel gains are regarded as primary and secondary users, respectively. The secondary user is only permitted to transmit on the condition that it would not impact the transmission of primary user adversely. Consequently, this setting would be sufficient to achieve significant difference in allocated powers of both users at the BS, which in turn facilitates the SIC operation at users' receivers. The problem of user pairing in uplink NOMA systems is studied in [72] for two network settings under fixed power allocation policies. The first scenario assumes that both BS and users are equipped with single antenna. Under this setting, a sub-optimum user pairing technique is proposed which is capable of achieving optimal performance with polynomial time complexity. In the second network setting, the BS is considered to have multiple antennas while users are equipped with single antenna. The results demonstrate that the optimal user pairing method achieves superior performance than the conventional random user pairing technique. Based on the differences in channel conditions of the users, the authors in [73] proposed user pairing and access theme algorithms. The results depict that the proposed schemes outperform the existing techniques in terms of capacity while ensuring the fairness among users. In [74], the problem of user pairing is addressed by dividing the users in two groups. A closed-form power allocation strategy is proposed in order to solve the problem of power allocation for NOMA users. The authors presented the results in terms of sum-rate maximization of NOMA system and demonstrated that the proposed user pairing method is capable of achieving optimal performance. The authors in [75] proposed an iterative algorithm in order to solve the joint optimization problem of user pairing and beamforming for NOMA systems. The results show that the proposed algorithm is capable of achieving superior rate fairness among NOMA users when compared to conventional random pairing and beamforming NOMA systems. In order to optimize the achievable rate, the problem of joint user pairing and power allocation is investigated in [76] under the condition of meeting minimum rate requirements of each user. The authors first obtain the optimized solutions by considering simpler systems with two and four users, and then derived the closed-form global optimal solution for generalized NOMA system. The results demonstrate that the proposed optimal scheme outperforms the conventional OMA as well as NOMA systems with random user pairing. An interesting problem of pairing mid-cell users who do not have significant difference in channel qualities is addressed in [77]. In order to solve this issue, the authors proposed two user pairing schemes which have a potential of enhancing capacity for all the users while treating the problem of pairing mid-cell users. In [78], the technique of PF is considered to achieve user pairing in the NOMA system. Considering the two user NOMA system, the authors derived a closed-form optimal solution

with PF objective in order to obtain user pairing and power allocation. Based on adaptive PF technique, the authors in [79] proposed an algorithm in order to obtain optimal user pairing, spectrum resource, and power allocation. The results show that the proposed algorithm achieves a comparable throughput performance to the global optimal search method and water-filling power allocation schemes, while also enhancing fairness among users. He et al. [80] proposed a novel fast user pairing algorithm which pairs users based on PF metric under fixed power allocation scheme. The simulation results demonstrate that at the expense of negligible throughput loss, the proposed user pairing method performs considerably faster than the conventional exhaustive search technique. The authors in [81] proposed an optimized scheme to jointly obtain user pairing and power allocation in order to maximize PF metric under the transmission power constraints. The simulation results illustrate a superior throughput performance and decreased complexity of the proposed scheme over current techniques in NOMA systems. In [82], the authors proposed a sort-based method in order to obtain user pairing and power allocation in NOMA system. The results depict that the proposed sort-based technique has a significant potential to enhance the error probability and spectral efficiency of downlink NOMA systems. The authors in [71] have investigated the problem of dynamic user clustering and power allocation by considering both uplink and downlink NOMA transmission scenarios. Based on transmit power, users' targeted rates, and SIC constraints, the authors proposed an optimal user clustering and power allocation method which maximizes the overall system throughput. Hsiung et al. [83] proposed a packet level scheduling scheme for BS that dynamically switches between NOMA or OMA. Further, the proposed scheduling method also obtains user pairing and power allocation for the considered NOMA system. The results demonstrate that the proposed scheme outperforms the distance-based user pairing technique in terms of throughput.

The issue of fairness for SCST NOMA networks has been studied in [84–87]. The work of Timotheou et al. [84] can be regarded as a pioneer study investigating the issue of fairness for NOMA. By considering instantaneous and average channel state information (CSI) at the transmitter, the authors proposed optimal power allocation for NOMA systems which guarantees fairness among downlink NOMA users. The authors in [85] considered the asymmetric channel and studied the fairness comparison between NOMA and OMA in terms of resource allocation. Based on Jain's index, the authors proposed a fairness indicator metric which facilitates the users' ability to select between NOMA and OMA schemes, and thus constitute a hybrid NOMA-OMA system. Simulation results demonstrate that the proposed fairness metric improves the fairness among users in hybrid NOMA-OMA system when compared to OMA and NOMA transmission techniques. In [86], the authors proposed a short term fairness scheme which aims to enhance the throughput–fairness trade-off for downlink NOMA systems. The results report an improvement of 38.96% by the proposed scheme in user fairness over conventional techniques. The authors in [87] suggested a new definition for user fairness which is based on the power distribution among different NOMA users. More precisely, under the proposed fairness index, the user's fair rate is determined according to the amount of its allocated power i.e., the user with larger power should obtain a higher rate. In order to confirm the precision of the proposed fairness index, the authors carried out an asymptotic analysis by considering both low and high SNR values. The problem of optimal resource allocation and energy efficiency for NOMA systems is investigated in [88–93] from the perspective of user fairness. In [88], the authors proposed an optimal power allocation scheme which aims to maximize overall energy efficiency of the uplink NOMA system while also accounting fairness among users. Based on an instantaneous CSI, the authors in [89] provide analytical expressions to obtain optimal power allocation in order to maximize the instantaneous sum-rate while maintaining the $\alpha$-fairness among users. By considering statistical and perfect CSI at the transmitter, the authors in [90] proposed an optimal power allocation algorithm which aims to maximize the overall throughput of NOMA systems with $\alpha$-fairness. Considering the identical user fairness level, the numerical results demonstrate that the NOMA under the proposed optimal power allocation method significantly enhances the system performance over conventional OMA technique. The authors in [91] considers a two-user downlink NOMA system over fading

channels and studied the trade-offs between system throughput and users' fairness. Based on full and partial CSI at the transmitter, the authors derived closed-form analytical expressions to obtain optimal power allocation policies for the considered NOMA system. The optimal fairness policy for NOMA systems is proposed in [93] which aims to maximize the achievable rate of the worst NOMA user. The authors derived a closed-form expression for the optimal fairness value and proposed an iterative algorithm in order to obtain the optimal solution with linear convergence. The trade-off between two conflicting metrics of sum-rate and fairness for multiple-input single-output (MISO) NOMA systems is further investigated in [92]. Based on different network requirements and channel conditions, the authors proposed a flexible beamforming design which is capable of catering both sum-rate and user fairness.

The aforementioned discussion reviews NOMA-related operations in the context of existing literature. Apart from NOMA specific issues, there are two promising technologies, namely MIMO and cooperative communications, which can be integrated with NOMA in order to further enhance the spectrum efficiency [25,27]. However, the main emphasis of this survey is to review the application of NOMA to different wireless networks and not to provide in depth survey on integrating different communication technologies to NOMA networks. Therefore, for completeness of discussion, in what follows, we will briefly review some state-of-the-art works on MIMO-NOMA and cooperative NOMA networks. The current literature e.g., see [94–98] explore the potentials of applying MIMO technique to NOMA networks. In [94,95], the authors investigated the capacity gain achieved by MIMO-NOMA over conventional MIMO-OMA systems. The authors derived closed-form analytical expressions for both sum channel and ergodic sum capacities to demonstrate the dominance of MIMO-NOMA over MIMO-OMA. Moreover, the authors in [96,97] performed system and link-level simulations to highlight the potentials of applying MIMO to NOMA. In addition, the authors developed hardware test-bed for NOMA, where SIC receiver was implemented in order to account the realistic hardware impairments. The results obtained after simulations and measurements demonstrate a 30% enhancement in cell throughput by NOMA over OFDMA system. The authors in [98] considered a downlink single user MIMO-NOMA system and evaluated the performance of cell-edge users by using system level simulations. Furthermore, a concept of signal alignment is applied to propose a general MIMO framework for both uplink and downlink NOMA systems in [99]. Considering the randomly deployed users and interferers, the authors derived a closed-form analytical expressions by using techniques from stochastic geometry in order to evaluate the performance of the considered system. Numerical results are presented in terms of outage probability which reveal the superior performance of MIMO-NOMA over conventional MIMO-OMA system. The benefit of applying cooperation in NOMA is investigated in [100–105]. The work of Ding et al. [100], can be regarded as a pioneer work which proposed a cooperative downlink NOMA system. The entire transmission is divided in two phases, namely, direct and cooperative, transmissions. The operating principle of direct transmission phase exactly follows the working of conventional NOMA system. The consequence of direct phase is that every user would have decoded its message as well as the messages of all the higher order users. As a result, the users broadcast decoded messages of higher order users by utilizing time slots in cooperative transmission phase. In order to evaluate the performance, the authors derived closed-form expressions for outage probability and diversity order for the proposed scheme. The simulation results validate the accuracy of the derived analytical expressions and also demonstrate the superior performance of the proposed NOMA-based network over a conventional cooperative OMA system. In [101], the authors proposed a two-stage relay selection strategy for cooperative NOMA networks and provide closed-form analytical expressions for outage probability. The numerical results reveal that the cooperative NOMA network under the proposed relay selection method is capable of achieving lower outage probability and better diversity gain than the conventional max–min approach for relay selection. Furthermore, the authors in [102] considered a cooperative NOMA system based on amplify-and-forward (AF) relay selection technique, and evaluated the performance of the considered system in terms of outage probability. The authors provided an approximation of the exact outage

probability and investigated the accuracy of the approximation by studying its asymptotic behavior. The results demonstrate that proposed scheme outperforms the conventional cooperative OMA system while maintaining the same diversity order. In addition, the authors in [103] considered a cooperative NOMA network and proposed a novel two-stage relay selection scheme based on decode-and-forward (DF) and AF relaying methods. Exact analytical expressions for outage probability are derived. Further, the authors performed asymptotic analysis in order to obtain diversity gain of the proposed scheme. The numerical results reveal that the proposed two-stage DF (AF) relaying for NOMA outperforms the existing relay selection methods for cooperative NOMA as well as OMA based networks. The authors in [104] considered a cooperative NOMA system in which relay possesses a buffer. Based on this relay feature, the authors proposed an adaptive transmission method which optimally selects the operational mode in every time slot. The results demonstrate that the proposed scheme significantly enhances the overall system throughput over conventional cooperative NOMA networks. In order to address the issues of high implementation complexity and power consumption of MIMO-NOMA systems, the authors in [105] proposed three node cooperative relaying system (CRS) based on spatial modulation (SM) aided NOMA (CRS-SM-NOMA) technique. Closed-form analytical results for bit-error-rate (BER) are provided in order to characterize the performance of the considered system. The numerical results demonstrate the superiority of the proposed CRS-SM-NOMA over CRS-NOMA and SM-OMA systems. The above discussion provides a brief and concise review for MIMO-NOMA and cooperative NOMA networks. Interested readers are referred to see [106–108] and [109–112] for more detailed treatment of MIMO-NOMA and cooperative NOMA, respectively.

*A. NOMA-Based SCST CN System*

Let us now shift our discussion on the working of SCST NOMA networks in this subsection. In NOMA-based SCST CN, there is a single BS communicating with multiple users by applying NOMA principle, as shown in Figure 5. At the user's receiver, SIC technique is applied in order to decode the message signal by mitigating the multiuser interference [27]. This arrangement constitutes a single-cell communication network, in which users are typically randomly distributed inside the coverage of BS. Considering total of $M$ NOMA users in a cell, the received signal at user $m$ can be obtained by using Equation (3). Consequently, the SINR available at user $m$ for decoding can be expressed by Equation (5).

*B. Outage Probability of NOMA-Based SCST CNs*

The user $m$ is considered to be in outage when its fails to detect the message of user $j$, where $1 \leq j \leq m$. This essentially implies that whenever user $m$ is unable to decode (meets the targeted rate) any of the higher order user or its own message, then it will suffer from complete outage. Consequently, based on Equation (5), the SINR at user $m$ to decode the message signal of $j$-th user, denoted by $\Gamma_{m \to j}$, can be expressed as:

$$\Gamma_{m \to j} = \frac{h_m a_j \rho}{h_m \rho \sum_{i=j+1}^{M} a_i + 1}. \tag{6}$$

Based on Equation (6), the outage probability at the $m$-th user in decoding $j$-th user message, denoted by $P_{m \to j}$ under an SCST NOMA-based CN can be expressed as follows:

$$
\begin{aligned}
P_{m \to j} &= \Pr \left( \Gamma_{m \to j} < \varphi_j \right) \\
&= \Pr \left( \frac{h_m a_j \rho}{h_m \rho \sum_{i=j+1}^{M} a_i + 1} < \varphi_j \right) \\
&= \Pr \left( h_m \rho \left( a_j - \varphi_j \sum_{i=j+1}^{M} a_i \right) < \varphi_j \right) \\
&= \Pr \left( h_m < \tau_j \right),
\end{aligned}
\tag{7}
$$

where $\varphi_j = 2^{R_j} - 1$, $R_j$ is the targeted rate of user $j$, $1 \leq j \leq m$, and $\tau_j = \frac{\varphi_j}{\rho \left( a_j - \varphi_j \sum_{i=j+1}^{M} a_i \right)}$. Note that the result in Equation (7) can only be obtained if the following condition is satisfied:

$$
a_j > \varphi_j \sum_{i=j+1}^{M} a_i.
\tag{8}
$$

It can be observed that the outage probability of user $m$ will always be one if condition in Equation (8) is not satisfied. This implies that careful allocation of rate and power would be required in order to keep the NOMA system operational. Next, let us define $\tau_m^{\max} = \max \{ \tau_1, ..., \tau_m \}$. The outage probability of the $m$-th user, denoted by $P_m^{\text{SCST}}$ can now be expressed as follows:

$$
\begin{aligned}
P_m^{\text{SCST}} &= \Pr \left( h_m < \tau_m^{\max} \right) \\
&= F_{h_m} \left( \tau_m^{\max} \right)
\end{aligned}
\tag{9}
$$

where $F_{h_m} (.)$ is the cumulative distribution function (CDF) of $h_m$. It can be observed from Equation (9) that in order to obtain outage probability of the $m$-th user in SCST NOMA CN, the CDF of channel gain $h_m$ would be required. Recall that channel gain $h_m$ in Equation (7) is given as $h_m = \frac{g_m}{1+d^\alpha}$, where $g_m$ is the fading power. Consider that $g_m$ has exponential distribution with unit mean $\sim \exp(1)$ i.e., assuming Rayleigh fading [113], then by applying order statistics [114], the CDF $F_{h_m}$ of channel gain can be expressed as follows [70,115]:

$$
F_{h_m} (x) = \mu_m \sum_{q=0}^{M-m} \binom{M-m}{q} \frac{(-1)^q}{m+q} \left( F_{\hat{h}} (x) \right)^{m+q},
\tag{10}
$$

where $\mu_m = \frac{M!}{(M-m)!(m-1)!}$, and $F_{\hat{h}} (x)$ is the CDF of the unordered channel gain. Next, consider that BS is located at the center of a disc with radius $\mathcal{R}_\mathcal{D}$, where $\mathcal{R}_\mathcal{D}$ models the coverage of BS. Further, since the users are randomly and uniformly distributed inside the coverage of BS and the fading is assumed to be Rayleigh, then, the CDF of the unordered channel gain is given as follows [116]:

$$
F_{\hat{h}} (x) = \frac{2}{\mathcal{R}_\mathcal{D}^2} \int_0^{\mathcal{R}_\mathcal{D}} \left( 1 - \exp \left( - (1 + r^\alpha) x \right) \right) r \, dr.
\tag{11}
$$

Note that finding the closed-form solution of Equation (11) for $\alpha > 2$ is very challenging. As such, the integral in Equation (11) can be approximated by Gaussian–Chebyshev quadrature in [117] or by utilizing confluent hyper-geometric function in [118]. Now, based on Equations (10) and (11), the outage probability of $m$-th user in SCST NOMA CN can be expressed in closed-form as follows:

$$P_m^{\text{SCST}} = \mu_m \sum_{q=0}^{M-m} \binom{M-m}{q} \frac{(-1)^q}{m+q} \left(F_{\hat{h}}\left(\tau_m^{\max}\right)\right)^{m+q}. \tag{12}$$

This completes the discussion of outage probability for SCST NOMA CN and we now describe the problem of optimal power allocation in next subsection.

*C. Sum-Rate Maximization Problem in NOMA-Based SCST CN*

The fundamental objective of any wireless system is to maximize the overall sum-rate of the network. Recall that NOMA operation is dependant on satisfying Equation (8) which would be met if rate and power allocations are done carefully. Consequently, an optimal power and rate allocations would be required in order to maximize the aggregated sum-rate of the NOMA-based SCST CN. In what follows, we will formulate and present a generic optimization problem that will maximize the overall sum-rate of the NOMA-based SCST CN subject to constraints accounting for practical aspects of the system. Let us define an objective function as, $\mathcal{F}^{\text{SCST}} = \sum_{m=1}^{M} \tilde{R}_m$, where $\tilde{R}_m = \log_2\left(1 + \Gamma_m\right)$ is the achievable rate of user $m$. Now, the constrained optimization problem can be formulated as follows:

$$\max \mathcal{F}^{\text{SCST}} \tag{13}$$

subject to:

$$\mathbf{C_1}: \quad R_m \leq \tilde{R}_m$$

$$\mathbf{C_2}: \quad \sum_{m=1}^{M} a_m P \leq P$$

$$\mathbf{C_3}: \quad \varphi_j \sum_{i=j+1}^{M} a_i \leq a_j, \quad \forall 1 \leq j \leq M.$$

The constraint $\mathbf{C_1}$ guarantees that the minimum rate requirements of all users are satisfied, which would also translate into minimum outage probability constraint for all the users. The condition $\mathbf{C_2}$ ensures that sum of powers allocated to all the users would ot exceed the total power budget $P$ available at the BS. Finally, the constraint $\mathbf{C_3}$ is added to appropriate users' rate and power allocation in order to keep NOMA with SIC operational. The above formulated problem is mixed-integer non-linear programming (MINLP). The solution of this MINLP problem has combinatorial nature. The given problem is hard to solve by utilizing standard optimization techniques. As such, one possible approach to solve this problem is to divide the problem into a sequence of linear programs and then develop an algorithm to obtain the optimal solution [84]. Another technique is to utilize standard optimization solvers such as CPLEX, MATLAB, and so on, to find the optimal solution numerically. Note that based on underlying modulation used in NOMA, the generic optimization presented in Equation (13) problem can be be easily tailored for SCST CN.

3.1.2. NOMA in Single-Cell Multi-Tier Cellular Networks

In current literature, NOMA-based single-cell multi-tier (SCMT) CNs are relatively less explored. The problem of resource allocation for NOMA-based single cell heterogeneous (multi-tier) networks is investigated in [119]. In the considered model, NOMA protocol is applied by only small-cell BSs (SBSs) to communicate with their users. In order to maximize the sum-rate of SBS users, the authors presented a resource allocation problem in terms of many-to-one matching game. The authors then proposed a novel distributed algorithm to obtain the solution of the formulated game. The results demonstrated the fast convergence of their proposed algorithm to a global optimal solution and SBS users achieve superior sum rate over OMA under the proposed resource allocation strategy. However, the SBSs are typically deployed in areas of higher user density, i.e., they usually follow user-centric

deployments in which the locations of users and SBS are correlated. Without considering this aspect, their modeling approach may not be accurate for modeling users' locations in heterogeneous networks with SBSs. Further, the authors in [120] considered a NOMA-based SCMT CN with single MBS and multiple SBSs, where SBSs are communicating with small-cell users via NOMA protocol. Considering the users' fairness issue, the authors proposed an optimal joint spectrum allocation and power control algorithm in order to maximize the sum-rate of the small-cell users. The results demonstrate that under the proposed algorithm, NOMA-based SCMT CN outperforms the conventional OMA based SCMT CN. Furthermore, the problem of resource allocation in energy-cooperation enabled two-tier CNs is investigated in [121], where MBSs and pico BSs (SBSs) apply the NOMA scheme to communicate with corresponding users. Moreover, in contrast to conventional grid power, the BSs in the considered system are also powered by renewable energy sources. In order to maximize the energy efficiency of such networks, the authors proposed an optimal algorithm to determine user association and power control while meeting the quality-of-service (QoS) requirements. The results reveal that the NOMA in SCMT CN outperforms the conventional OMA in the considered network in terms of energy efficiency. The problem of optimal power allocation is considered in [122] under imperfect CSI in order to maximize the energy efficiency of the small-cells. Based on the Lagrangian method, closed-form optimal power allocation for each small-cell is derived. The results demonstrate that the considered SCMT network under the proposed scheme outperforms the conventional algorithms. The issue of offloading MBS users in an SCMT scenario under congested situations is investigated in [123]. In order to serve the offloaded MBS users, FBS tier applies user pairing and then utilizes the NOMA protocol for serving the users. With the proposed offloading strategy, the authors have reported a significant improvement of 74.04% and 48.65% in outage probability for cell center and cell-edge users, respectively. Furthermore, the authors considered a single MBS cell underlaid with femto BSs (FBSs) cells. In [124], the authors considered a SCMT scenario, where SBSs (femto BSs in [124]) and D2D tiers are underlaid with single MBS. NOMA transmission protocol is applied by SBS and D2D tiers. The authors investigated the problem of congestion in the considered network and suggested to offload MBS load to SBSs. Further, when SBSs are unable to pair the offloaded MBS users, the authors proposed to utilize the D2D tier in order to support these unpaired offloaded MBS users. The results show that the proposed offloading scheme enormously decreases the outage probability and enhances the ergodic rate of offloaded MBS users. Considering an ultra dense heterogeneous (SCMT) network, the authors in [125] proposed an optimal fair power allocation scheme which aims to maximize the overall energy efficiency of the considered network. The simulation results are presented in terms of energy efficiency for the considered network which demonstrate superior performance over conventional OMA based SCMT CNs. The problems of energy and spectrum efficiencies are considered in [126] for NOMA-based SCMT CN. The constrained multi-objective optimization problem is formulated in order to maximize the overall energy and spectrum efficiency of the network. The authors proposed novel optimal algorithms to allocate power and sub-channels and reported superior performance of the considered NOMA-based CN over conventional OMA. The authors investigated the problem of user scheduling and power optimization in [127] and studied the trade-off between data rate and energy consumption for NOMA systems. By considering perfect and imperfect CSI, the authors proposed user scheduling and power allocation schemes. Under the proposed resource allocation methods, the results reveal significant enhancement in performance of NOMA-based SCMT network for both perfect and imperfect CSI scenarios. Based on minimum user rate requirements, the authors in [128] considered a NOMA-based SCMT CN in which both MBS and SBS compete in order to maximize their throughputs. The power allocation required to optimize the MBS and SBS individual throughputs is modeled as a Stackelberg game. In order to obtain the Stackelberg equilibrium, the author proposed a distributed power allocation algorithm which has improved the spectrum efficiency of the considered NOMA-based SCMT CN. Furthermore, the authors in [129] investigated the problem of subchannel and power allocation in order to maximize the energy efficiency for NOMA-based SCMT CN. The optimal solution for subchannel and power allocation is derived in closed-form and the results show the fast

convergence of the proposed optimal algorithm while significantly enhancing overall system energy efficiency. Moreover, The authors in [130] proposed an optimal joint subchannel and power allocation algorithm which is aimed to enhance the energy efficiency of MBS and SBS in a NOMA-based SCMT CN. Numerical results reveal that the proposed algorithm is capable of improving energy efficiency over the existing schemes. The problem of joint spectral and energy efficiency is investigated by the authors in [131] for NOMA-based SCMT CN. The results demonstrate that the proposed scheme achieve superior performance in terms of spectral and energy efficiency over conventional NOMA and OMA. Considering NOMA-based SCMT CN, the authors in [132] investigated the trade-off among four quantities, namely, sum-rate, fairness, energy efficiency, and harvested energy. The authors proposed an optimal algorithm in order to jointly allocate subcarrier and power which aims to obtain energy efficient resource allocation. Furthermore, considering two scenarios of resource allocation based on subcarrier and intercell interference, the authors in [133] proposed an optimal algorithm to maximize the sum-rate of the entire system. The results demonstrate that the system link performance is enhanced under the proposed scheme when the system sum-rate is kept low. The authors in [134] studied the average rate, coverage probability, and quality of experience (QoE) for NOMA-based SCMT CN. The results demonstrate that the considered network achieves better rate and QoE compared to the conventional OMA. By considering the NOMA technique implemented by both MBS and SBS in SCMT CN, the authors in [135] proposed an optimal power allocation algorithm in order to maximize the overall system throughput. The results demonstrate that the considered network under the proposed power allocation algorithm significantly enhances the spectral efficiency and improves the outage performance over the conventional OMA. The authors in [136] considered a ratio fairness model and user scheduling scheme in order to increase the overall system capacity of the NOMA-based SCMT CN. In addition, a user pairing method is also suggested to improve the working of existing SIC. The results demonstrate that the proposed scheme achieves superior fairness performance compared to the existing NOMA fairness scheme and conventional OMA. The authors in [137] consider a NOMA-based SCMT CN with self-organizing network (SON) functionalities. In particular, the authors have proposed a distributed algorithm in which under situations of congestion, the users from congested tiers are offloaded to achieve load balancing while the resource allocation is obtained such that it would minimize the co-channel interference. The simulation results reveal that the improvement in data rate and user fairness has been observed by considering SON functionality with NOMA-based SCMT CN.

*A. NOMA-Based SCMT CN System*

A general NOMA-based *K*-tier single-cell CN is depicted in Figure 6. It can be observed that there is a single macro-cell in which MBS is located at the center of a disc with radius $\mathcal{R}_\mathcal{D}$, where $\mathcal{R}_\mathcal{D}$ models the coverage of MBS. Without a loss of generality, it can be assumed that the MBS forms tier one. Consequently, the BSs from remaining $K-1$ tiers constitute small-cells, and are referred as SBSs. Further, it can be assumed that each *k*-th tier SBS is located at the center of a disc with radius $\mathcal{R}_{\mathcal{D},k}$, where $\mathcal{R}_{\mathcal{D},k}$ represents the coverage of the *k*-th tier SBS and $1 < k \leq K$. Further, we call the users who fall inside coverage of any SBS as clustered users (CUs), while those who do not fall inside coverage of any SBS are named as non-clustered users (NCUs). Note that at particular time instant, a NCU is always served by a MBS, whereas CU could be associated to MBS or SBS. As such, let us call the users who are served by MBS as MBS users (MUs), while those served by SBS are termed as SBS users (SUs). Note that SUs are always CUs whereas MUs can be further classified as non-clustered MUs (NCMUs) and clustered MUs (CMUs) [138]. Further, for analysis purposes, we consider a representative *k*-th tier small-cell in which *N* SUs are randomly distributed. In addition, consider that there are total *M* MUs which are distributed randomly inside coverage of MBS. Moreover, assume that $\{h_m\}_{m=1}^{M}$ and $\left\{h_n^k\right\}_{n=1}^{N}$ represent the channel gains (power) for *m*-th MU and *n*-th SU in *k*-th tier, respectively. Consequently, based on the NOMA principle, the MUs and SUs are considered to be ordered as $h_1 \leq .... \leq h_M$ and

$h_1^k \leq .... \leq h_N^k$, respectively, where $h_m = \frac{g_m}{d_m^\alpha}$ and $h_n^k = \frac{g_n^k}{\{d_n^k\}^\alpha}$. Here, $g_n^k$ and $d_n^k$ represent the fading power and distance between the $n$-th user and $k$-th tier SBS. As a result, power allocation coefficients for MUs and SUs in $k$-th tier are sorted as $a_1 \geq ... \geq a_M$ and $a_1^k \geq ... \geq a_N^k$, respectively, where $a_n^k$ is the power allocation coefficient of $n$-th SU in tier $k$. Further, consider that all BSs are utilizing the same time and frequency resources. Then, the received signal power at $m$-th MU, denoted as $P_m^{\text{SCMT}}$ can be expressed as follows:

$$P_m^{\text{SCMT}} = \begin{cases} \underbrace{h_m a_m P}_{\text{useful signal power}} + \overbrace{h_m \sum_{\substack{i=1 \\ i \neq m}}^{M} a_i P}^{\text{intra-user interference}} + \sigma^2, & \text{if } m\text{-th user is NCU} \\[3em] \underbrace{h_m a_m P}_{\text{useful signal power}} + \underbrace{h_m \sum_{\substack{i=1 \\ i \neq m}}^{M} a_i P}_{\text{intra-user interference}} + \mathcal{I}_m^{\text{SBS}} + \sigma^2, & \text{if } m\text{-th user is CU} \end{cases} \tag{14}$$

where $\mathcal{I}_m^{\text{SBS}}$ is the total received interference (power) at $m$-th MU from SBSs. Similarly, the received power at the $n$-th SU in tier $k$, denoted by $P_n^{k,\text{SCMT}}$ can be expressed as follows:

$$P_n^{k,\text{SCMT}} = \underbrace{h_n^k a_n^k P_k}_{\text{useful signal power}} + \overbrace{h_n^k \sum_{\substack{i=1 \\ i \neq n}}^{N} a_i^k P_k}^{\text{intra-user interference}} + \mathcal{I}_n^{k,\text{MBS}} + \mathcal{I}_n^{k,\text{SBS}} + \sigma^2, \tag{15}$$

where $\mathcal{I}_n^{k,\text{MBS}}$ and $\mathcal{I}_n^{k,\text{SBS}}$ represent the total interference at $n$-th SU in tier $k$ from MBS and other SBSs, respectively.

Note: The exact characterization of interference terms $\mathcal{I}_m^{\text{SBS}}$ and $\mathcal{I}_n^{k,\text{MBS}}, \mathcal{I}_n^{k,\text{SBS}}$ in Equations (14) and (15), respectively, and all the interference terms that will arise in subsequent discussions, require stochastic geometry based modeling of BSs and user distributions, which is beyond the scope of this survey. Interested readers are referred to see [139–144] for more detailed treatment of this topic.

*B. Outage Probability of NOMA-Based SCMT CN*

Recall that NCU always connects to MBS whereas CU can be connected to MBS or SBS. As such, let us denote by $\mathcal{A}^{\text{MBS}}$ and $1 - \mathcal{A}^{\text{MBS}}$ as the probability that CU is associated to MBS and SBS, respectively. The probability of association for an arbitrary CU to connect to MBS can be expressed as follows:

$$\begin{aligned} \mathcal{A}^{\text{MBS}} &= \text{P}\left( \text{R}_{\text{MBS}}^{-\alpha} P > \text{R}_{\text{SBS}}^{-\alpha} P_k \right) \\ &= \text{P}\left( \text{R}_{\text{MBS}} < \left( \frac{P}{P_k} \right)^{\frac{1}{\alpha}} \text{R}_{\text{SBS}} \right) \\ &= \int_0^\infty \left[ 1 - F_{\text{R}_{SBS}}\left( \left( \frac{P}{P_k} \right)^{\frac{1}{\alpha}} r_{\text{MBS}} \right) \right] f_{\text{R}_{\text{MBS}}}\left( r_{\text{MBS}} \right) dr_{\text{MBS}}, \end{aligned} \tag{16}$$

where $\text{R}_{\text{MBS}}$ and $\text{R}_{\text{SBS}}$ represent the distance between a typical CU and the nearest MBS and SBS, respectively. It can be observed from Equation (16) that the association probability $\mathcal{A}^{\text{MBS}}$ depends on the distributions of random variables $\text{R}_{\text{MBS}}$ and $\text{R}_{\text{SBS}}$. In current literature, one promising method to determine these distributions is to utilize a stochastic geometry framework, e.g., see the references [145,146]. Therefore, after obtaining $F_{r_{\text{SBS}}}$ and $f_{r_{\text{MBS}}}$ and solving the integral in Equation (16) provides the association probability $\mathcal{A}^{\text{MBS}}$.

Next, we focus our discussion on expressing the outage probability for typical MU and SU in SCMT CN. Based on Equation (14), the received SINR at $m$-th MU after removing higher order users, denoted by $\Gamma_m^{\text{SCMT}}$, can be expressed as follows:

$$\Gamma_m^{\text{SCMT}} = \begin{cases} \dfrac{h_m a_m \rho}{h_m \rho \sum_{i=m+1}^{M} a_i + 1}, & \text{if } m\text{-th user is NCU} \\[2ex] \dfrac{h_m a_m \rho}{h_m \rho \sum_{i=m+1}^{M} a_i + \bar{\rho} \mathcal{I}_m^{\text{SBS}} + 1}, & \text{if } m\text{-th user is CU} \end{cases} \tag{17}$$

where $\bar{\rho} = \frac{1}{\sigma^2}$. Now, at a particular time instant, MU can be of non-clustered or clustered type (but can't be of both types). As such, let us define the following events in order to determine the outage probability of the $m$-th MU under SCMT CN.

$\varepsilon_1 = \{\text{Tpical MU is NCU}\}$
$\varepsilon_2 = \{\text{Tpical MU is CU}\}$
$\varepsilon_3 = \{\text{Outage at tpical MU when it is NCU in SCMT CN}\}$
$\varepsilon_4 = \{\text{Outage at tpical MU when it is CU in SCMT CN}\}$

Based on the events $\varepsilon_1 - \varepsilon_4$, the outage probability at the $m$th MU, denoted by $P_m^{\text{SCMT}}$, can be expressed as follows:

$$P_m^{\text{SCMT}} = P_{\varepsilon_1} P_{\varepsilon_3} + P_{\varepsilon_2} P_{\varepsilon_4}. \tag{18}$$

Now following similar steps to deriving Equation (12), the probability $P_{\varepsilon_3}$ can be expressed as follows:

$$P_{\varepsilon_3} = P_m^{\text{SCST}}. \tag{19}$$

Next, condition on interference $\mathcal{I}_m^{\text{SBS}}$ and following similar steps to Equation (12), the probability $P_{\varepsilon_4}$ can be written as:

$$\begin{aligned} P_{\varepsilon_4} &= P_{\varepsilon_4 | \mathcal{I}_m^{\text{SBS}}} \\ &\overset{(a)}{=} \int_0^{\infty} P_m^{\text{SCST}} \big|_{\tau_m^{\max} = \bar{\tau}_m^{\max}} f_{\mathcal{I}_m^{\text{SBS}}} (x) \, dx, \end{aligned} \tag{20}$$

where $\bar{\tau}_m^{\max} = \max \{\bar{\tau}_1, ..., \bar{\tau}_j, ...., \bar{\tau}_m\}$, $\bar{\tau}_j = \dfrac{\bar{\varphi}_j (1 + \bar{\rho} \mathcal{I}_m^{\text{SBS}})}{\rho \left(a_j - \bar{\varphi}_j \sum_{i=j+1}^{M} a_i\right)}$, $\bar{\varphi}_j = 2^{R_j^{\text{SCMT}}} - 1$, $1 \le j \le m$, $f_{\mathcal{I}_m^{\text{SBS}}} (x)$ is the PDF of interference $\mathcal{I}_m^{\text{SBS}}$ and $(a)$ in Equation (20) is obtained by de-conditioning on $f_{\mathcal{I}_m^{\text{SBS}}} (x)$. Finally, using Equations (19) and (20) in Equation (18) derives the outage probability expression for $m$-th MU in SCMT NOMA CN.

Next, recall that at some particular time instant, SU can only be connected to SBS. As such, based on Equation (15), the SINR at typical $k$-th tier $n$-th SU, denoted by $\Gamma_n^{k,\text{SCMT}}$, can be expressed as follows:

$$\Gamma_n^{k,\text{SCMT}} = \frac{h_n^k a_n^k \rho_k}{h_n^k \rho_k \sum_{i=n+1}^{N} a_i^k + \bar{\rho} \left(\mathcal{I}_n^{k,\text{MBS}} + \mathcal{I}_n^{k,\text{SBS}}\right) + 1}, \tag{21}$$

where $\rho_k = \frac{P_k}{\sigma^2}$. Now following similar approach in deriving Equation (20), the outage probability the typical $k$-th tier $n$-th SU, denoted by $P_n^{k,\text{SCMT}}$, can be written as:

$$P_n^{k,\text{SCMT}} = \int_0^{\infty} \int_0^{\infty} P_m^{\text{SCST}} \big|_{\tau_m^{\max} = \bar{\tau}_{n,k}^{\max}} f_{\mathcal{I}_n^{k,\text{MBS}}} (x) \, f_{\mathcal{I}_n^{k,\text{SBS}}} (y) \, dxdy, \tag{22}$$

where $\bar{\tau}_{n,k}^{\max} = \max\left\{\bar{\tau}_{1,k}, ..., \bar{\tau}_{j,k}, ...., \bar{\tau}_{n,k}\right\}$, $\bar{\tau}_{j,k} = \frac{\check{\varphi}_{j,k}(1+\bar{\rho}(x+y))}{\rho_k\left(a_j^k - \check{\varphi}_j^k \sum_{i=j+1}^{N} a_i^k\right)}$, $\check{\varphi}_j = 2^{R_j^{k,\text{SCMT}}} - 1$, $1 \le j \le n$, $R_j^k$ is

the targeted rate for $j$-th SU in tier $k$, $f_{\mathcal{I}_n^{k,\text{MBS}}}(x)$, and $f_{\mathcal{I}_n^{k,\text{SBS}}}(y)$ are the PDFs of interference $\mathcal{I}_n^{k,\text{MBS}}$ and $\mathcal{I}_n^{k,\text{SBS}}$, respectively. This completes the discussion for outage probability in NOMA-based SCMT CNs.

*C. Sum-Rate Maximization Problem in NOMA-Based SCMT CN*

Similar to SCST CN case, let us consider a problem of maximizing overall sum-rate in SCMT NOMA-based CN. As such, define an objective function as follows:

$$\mathcal{F}^{\text{SCMT}} = \sum_{m=1}^{M} \tilde{R}_m^{\text{SCMT}} + \sum_{k=2}^{K} \sum_{n=1}^{N} \tilde{R}_n^{k,\text{SCMT}}, \tag{23}$$

where $\tilde{R}_m^{\text{SCMT}}$ and $\tilde{R}_n^{k,\text{SCMT}}$ are the achievable rates of typical $m$-th MU and $k$-th tier $n$-th SU in SCMT CN. Now, based on Equation (23), the optimization problem can be formulated as follows:

$$\max \mathcal{F}^{\text{SCMT}} \tag{24}$$

subject to:

$\mathbf{C_1}:\quad R_m \le \tilde{R}_m^{\text{SCMT}}$

$\mathbf{C_2}:\quad R_n^k \le \tilde{R}_n^{k,\text{SCMT}}$

$\mathbf{C_3}:\quad \sum_{m=1}^{M} a_m P \le P$

$\mathbf{C_4}:\quad \sum_{n=1}^{N} a_n^k P_k \le P_k, \quad 2 \le k \le K$

$\mathbf{C_5}:\quad \varphi_j \sum_{i=j+1}^{M} a_i \le a_j, \quad 1 \le j \le M$

$\mathbf{C_6}:\quad \check{\varphi}_j^k \sum_{i=j+1}^{N} a_i^k \le a_j^k, \quad 1 \le j \le N$

$\mathbf{C_7}:\quad \mathcal{I}_m^{\text{SBS}} < h_m a_m P, \quad 1 \le m \le M$

$\mathbf{C_8}:\quad \mathcal{I}_n^{k,\text{MBS}} + \mathcal{I}_n^{k,\text{SBS}} < h_n^k a_n^k P_k, \quad 1 \le n \le M, 2 \le k \le K.$

The constraints $\mathbf{C_1}, \mathbf{C_2}$ guarantee that the minimum rate requirements for MUs and SUs are met. Further, the constraints $\mathbf{C_3}, \mathbf{C_4}$ ensure that the allocated powers to MUs and SUs would not exceed the total transmitted powers available at MBS and SBS. The necessary conditions required to keep NOMA-SIC operational are imposed by the constraints $\mathbf{C_5}, \mathbf{C_6}$. Finally, the constraints $\mathbf{C_7}, \mathbf{C_8}$ are imposed to ensure that the MUs and SUs are allocated with appropriate power levels such that the received powers at SUs and MUs should exceed corresponding interference levels.

### 3.1.3. NOMA in Multi-Cell Single-Tier Cellular Network

Much less attention has been paid in existing literature to investigating NOMA-based multi-cell single-tier (MCST) CNs. The authors in [147] analyze the uplink NOMA for large-scale cellular networks. The spatial locations of the BSs are modeled by a homogeneous PPP. Considering the locations of BSs as a parent PP, the spatial topology of users is modeled by PCP, in which users are clustered around the locations of BSs. This captures the correlation between the location of users and BS. In particular, they considered three scenarios of SIC, namely perfect SIC, imperfect SIC, and imperfect worst case SIC. In order to evaluate the performance of the considered network, the authors first derived a closed-form expression for intra-cluster interference. Then, based on the interference results, an expression for rate coverage is provided under all three scenarios of SIC. Numerical simulations are

conducted to validate the accuracy of the derived analytical results and compare the proposed network with PPP- and OMA-based networks. The results show that the PCP-based network outperforms the OMA-based network and offers more accurate analysis than the PPP-based approach, which provides optimistic results for NOMA. However, the considered system model is limited to a single tier case.

The problem of power allocation for OFDM based NOMA systems is studied in [148]. The considered cellular network is composed of 19 hexagonal grid sites having three cells per site. The authors formulated an optimization problem to maximize the weighted sum rate of the entire system. Due to the intractability of the presented optimization problem, it was solved in two steps. In the first step, the user selection across each subchannel and power assignment are performed by using greedy, and sub-optimal iterative power allocation algorithms, respectively. In the second step, using the iterative power allocation algorithm of the first part, the power assignment across the sub-channel is achieved. The simulation results demonstrate the superiority of their proposed scheme over existing ones in terms of achievable system throughput. However, a fundamental limitation of their system model is the assumption of hexagonal grids with a regular shape and size, which may not always hold true in real cell site deployments.

A NOMA-based multi-cell network is analyzed in [149]. The locations of BSs are modeled by a homogeneous PPP, and a more realistic Voronoi cell structure is considered. The authors provided close-form expressions for the coverage probability and achievable rate. Numerical simulations are conducted to validate the accuracy of derived analytical results, and compare the achieved performance with that of an OMA-based network. The results show that NOMA achieves higher coverage and achievable rate than OMA under the considered network setting. However, there are two potential shortcomings: (1) The performed analysis is limited to a case of only two users for single-tier networks; and (2) modeling locations of MBSs by PPP is not accurate because real deployments of MBSs exhibit interaction between MBSs locations, whereas PPP does not capture this property due to its completely spatial random property. The key challenges and benefits of applying NOMA in a MCST environment is discussed in [150]. In addition, the authors have also comprehensively discussed the theory and working of NOMA in MCST and SCST cases.

*A. NOMA-Based MCST CN System*

Figure 7 illustrates the basic configuration of MCST CN utilizing NOMA protocol. Similar to SCST and SCMT CNs, it can be observed from the Figure 7 that each MBS is located at the center of a disc with radius $\mathcal{R}_D$ modeling the coverage of MBS in MCST CN. The MBSs are communicating with users by applying NOMA protocol. Further, consider that there are total of $M$ NOMA users distributed randomly in each macro-cell. For analysis purposes, we consider $m$-th user as typical MU in a representative macro-cell. The situation of $m$-th NOMA user in a representative macro-cell is similar to SCST scenario except that there would be an additional interference from other MBSs in the considered MCST CN. Therefore, the received power at the $m$-th user in a representative macro-cell, denoted by $P_m^{MCST}$, can be written as follows:

$$P_m^{\text{MCST}} = \underbrace{h_m a_m P}_{\text{useful signal power}} + \underbrace{h_m \sum_{\substack{i=1 \\ i \neq m}}^{M} a_i P}_{\text{intra-user interference}} + \mathcal{I}_m^{\text{MBS}} + \sigma^2, \tag{25}$$

where $\mathcal{I}_m^{\text{MBS}}$ is the interference at the typical $m$-th MU in a representative macro-cell from other MBSs in MCST CN.

*B. Outage Probability of NOMA-Based MCST CN*

Based on Equation (25), the SINR at $m$-th MU after the application of SIC, denoted as $\Gamma_m^{\text{MCST}}$, can be expressed as follows:

$$\Gamma_m^{\text{MCST}} = \frac{h_m a_m \rho}{h_m \rho \sum_{i=m+1}^{M} a_i + \bar{\rho} \mathcal{I}_m^{\text{MBS}} + 1}. \tag{26}$$

Next, following the similar approach as was used in obtaining Equation (20), the outage probability at $m$-th MU in MCST CN, denoted as $\text{P}_m^{\text{MCST}}$ can de written as:

$$\text{P}_m^{\text{MCST}} = \int_0^{\infty} \text{P}_m^{\text{SCST}} \big|_{\tau_m^{\max} = \hat{\tau}_m^{\max}} f_{\mathcal{I}_m^{\text{MBS}}}(x)\, dx, \tag{27}$$

where $\hat{\tau}_m^{\max} = \max\left\{\hat{\tau}_1, ..., \hat{\tau}_j, ...., \hat{\tau}_m\right\}$, $\hat{\tau}_j = \dfrac{\hat{\varphi}_j\left(1 + \bar{\rho}\mathcal{I}_m^{\text{MBS}}\right)}{\rho\left(a_j - \hat{\varphi}_j \sum_{i=j+1}^{M} a_i\right)}$, $\hat{\varphi}_j = 2^{R_j^{\text{MCST}}} - 1$, $1 \leq j \leq m$, and $f_{\mathcal{I}_m^{\text{MBS}}}(x)$ is the PDF of interference $\mathcal{I}_m^{\text{MBS}}$.

*C. Sum-Rate Maximization Problem in NOMA-Based MCST CN*

Let us consider an objective function $\mathcal{F}^{\text{MCST}}$ for maximizing sum-rate of the representative macro-cell in MCST CN. Note that this is identical of maximizing the overall sum-rate of the MCST CN in terms of average performance. The objective function $\mathcal{F}^{\text{MCST}}$ can be now defined as:

$$
\begin{aligned}
\mathcal{F}^{\text{MCST}} &= \sum_{m=1}^{M} \tilde{R}_m^{\text{MCST}} \\
&= \sum_{m=1}^{M} \log_2\left(1 + \Gamma_m^{\text{MCST}}\right) \\
&= \sum_{m=1}^{M} \log_2\left(1 + \frac{h_m a_m \rho}{h_m \rho \sum_{i=m+1}^{M} a_i + \bar{\rho}\mathcal{I}_m^{\text{MBS}} + 1}\right).
\end{aligned} \tag{28}
$$

Note that objective function $\mathcal{F}^{\text{MCST}}$ in Equation (28) is different from $\mathcal{F}^{SCST}$ in Equation (13) due to the presence of interference term $\mathcal{I}_m^{\text{MBS}}$ in the achievable rate of user $m, 1 \leq m \leq M$ in MCST CN. The optimization problem can now be formulated as follows:

$$\max \mathcal{F}^{\text{MCST}} \tag{29}$$

subject to:

$$\mathbf{C_1}: \quad R_m^{\text{MCST}} \leq \tilde{R}_m$$

$$\mathbf{C_2}: \quad \sum_{m=1}^{M} a_m P \leq P$$

$$\mathbf{C_3}: \quad \hat{\varphi}_j \sum_{i=j+1}^{M} a_i \leq a_j, \quad 1 \leq j \leq M$$

$$\mathbf{C_4}: \quad \mathcal{I}_m^{\text{MBS}} < h_m a_m P, \quad 1 \leq m \leq M.$$

Note that compared to optimization problem in Equation (13), there is an additional constraint $\mathbf{C_4}$ in Equation (29) which ensures that by virtue of appropriate power allocation at MBS, each MU would received a power level that could exceed the corresponding interference level.

3.1.4. NOMA in Multi-Cell Multi-Tier Cellular Networks

The current literature is scarce in analyzing NOMA-based multi-cell multi-tier (MCMT) CNs. In [151], the authors consider the application of NOMA to K-tier multi-cell network. A comprehensive hybrid transmission framework is proposed under which MBSs and SBSs utilize massive MIMO and

NOMA technologies to enhance the overall spectral efficiency. Under the considered K-tier network, the authors proposed a BS association policy for MBS and SBS users based on the biased average received power metric. The performance of the network is evaluated in terms of spectrum efficiency achieved by each tier, where the closed-form expressions are derived by using stochastic geometry tools. Numerical results demonstrate that the proposed hybrid massive MIMO for NOMA-based multi-tier network can achieve better spectrum efficiency then its OMA counterpart. Its potential shortcoming is the use of less realistic user distributions that do not capture the coupling between users' and SBS locations. In addition, MBS locations are modeled by homogeneous PPP, which does not reflect the deployment trend of real MBSs.

The application of NOMA to multimedia broadcast/multicast service (MBMS) is investigated for multi-cell K-tier network in [152]. Such a network aims to meet the high data rate demands of emerging applications by enjoying the efficient spectrum utilization of both NOMA and MBMS techniques. The BS spatial topology of each tier is modeled by independent homogeneous PPP. Based on NOMA and MBMS, the authors proposed two transmission schemes and evaluated the performance of these schemes using stochastic geometry. An analytical framework is developed which is general enough to cover the cases of synchronous and asynchronous non-orthogonal MBMS transmission. Based on this framework, the authors derived closed-form expressions for coverage probability, sum rate and the number of users served. Simulations are conducted to verify the analysis. It was shown that non-orthogonal MBMS outperforms orthogonal MBMS. In addition, synchronous transmission mode can achieve better performance than the asynchronous one. However, the users in each tier are modeled by independent homogeneous PPPs. While this distribution is accurate for users who are located independent of BS locations, it does not capture the location coupling between users and SBS of multi-tier networks, where the latter is often deployed to enhance the service in areas of high congestion.

In [115], the authors considered a large-scale NOMA-based multi-tier cellular network in which SBSs communicate with their users with the aid of NOMA protocol, whereas MBSs are equipped with massive MIMO capability. A tractable analytical framework is developed to analyze the performance of considered network. Closed-form expressions for coverage probability are obtained for NOMA SBSs users. In addition, a lower bound for achievable ergodic rate is derived for a user served by massive MIMO enabled MBSs. Based on the coverage results, the energy efficiency of hybrid multi-tier networks is also investigated. Numerical results reveal that the performance of SBSs NOMA users is highly dependent on the choice of user targeted rates and power allocation coefficients. Moreover, equipping MBSs with massive MIMO capability significantly enhances the network's spectrum efficiency. While the developed model is analytically tractable, it suffers from two shortcomings: (i) NOMA is not utilized for MBSs, and (ii) it uses homogeneous PPPs to model MBSs and SBSs. The authors in [153] considered a NOMA-based MCMT CN and investigated the system performance in terms of coverage and throughput under non-coordinated and coordinated joint NOMA transmission techniques. The authors derived a constraint on power allocation of users which mist be satisfied in order to enhance the coverage and throughput of the considered system. In addition, the authors have also showed the existence of optimal power allocation that would maximize the coverage and throughput of the system. The authors in [154] considered a NOMA-based MCMT CN for ultra dense network deployments and proposed a network architecture which integrates ground–air–space radio access network and core networks by applying the concepts of virtualization. Moreover, the authors also studied the impact of fog-computing coordination on resource allocation and macro-small cell coordination. Simulation results demonstrate that utilizing network coordination would significantly enhance the system performance of both NOMA and OMA based MCMT ultra dense CNs. The authors in [155] considered a NOMA-based MCMT CN where small-cells are deployed in a non-uniform manner. In order to evaluate the system performance, the authors derived closed-form expressions for coverage probability and achievable rate. Numerical results are presented to validate the accuracy of the performed analysis and also demonstrate the superiority of NOMA over OMA under the

considered MCMT CN. Furthermore, the authors in [156] considered a NOMA-based multi-tier CN with cloud-based central station coordination functionality. By considering realistic power consumption of different cell types and wireless backhaul, the authors proposed a power allocation method for multiple cells from different tiers in the considered network. The authors reported that the overall energy efficiency of the network is hugely impacted by the power available at the cloud, cell types, and propagation environment. In [157], the authors have proposed a framework to analyze ultra dense NOMA-based MCMT CN. The proposed framework can be modified and configured easily to accommodate various application scenarios. Further, in order to highlight the effectiveness of their proposed framework, the authors have presented two case studies.

*A. NOMA-Based MCMT CN System*

Figure 8 shows a MCMT CN in which BS from each tier implements NOMA in order to communicate with the cellular users. Let us consider that there are total of *K* tiers, where without loss of generality, it is assumed that the tier one is comprised of MBSs. It can be observed that each MBS and *k*-th tier SBS are located at the center of discs with radii $\mathcal{R}_\mathcal{D}$ and $\mathcal{R}_{k,\mathcal{D}}$, $1 < k \leq K$. Further, similar to the SCMT case, for analysis purposes, consider a representative macro-cell in which multiple small-cells are underlaid. Note that in average terms, the performance of each macro-cell in MCMT CN is similar to the performance of representative macro-cell. As such, with this description and assumption for analysis, the representative macro-cell in the considered MCMT CN is similar to SCMT case with exception of interference at a typical MU and SU from other MBSs and all MBSs, respectively. Consequently, all the underlying user classifications (CMUs, NCMUs, etc.) and definition/notations defined earlier in SCMT CN system describing different variables (channel gains, power allocation, etc.) will remain same in the subsequent discussion. Then, the received power at *m*-th MU in MCMT CN, denoted by $P_m^{\text{MCMT}}$, can be expressed as follows:

$$
P_m^{\text{MCMT}} =
\begin{cases}
\underbrace{h_m a_m P}_{\text{useful signal power}} + \overbrace{h_m \sum_{\substack{i=1 \\ i \neq m}}^{M} a_i P}^{\text{intra-user interference}} + \mathcal{I}_{m,\text{MCMT}}^{\text{MBS}} + \sigma^2, & \text{if } m\text{-th user is NCU} \\[2em]
\underbrace{h_m a_m P}_{\text{useful signal power}} + \underbrace{h_m \sum_{\substack{i=1 \\ i \neq m}}^{M} a_i P}_{\text{intra-user interference}} + \mathcal{I}_{m,\text{MCMT}}^{\text{MBS}} + \mathcal{I}_{m,\text{MCMT}}^{\text{SBS}} + \sigma^2, & \text{if } m\text{-th user is CU}
\end{cases}
\tag{30}
$$

where $\mathcal{I}_{m,\text{MCMT}}^{\text{MBS}}$ and $\mathcal{I}_{m,\text{MCMT}}^{\text{SBS}}$ represent the interference at typical *m*-th MU from other MBSs and SBSs, respectively. Similarly, the received power at the typical *k*-th tier *n*-th SU, denoted by $P_n^{k,\text{MCMT}}$, can be expressed as follows:

$$
P_n^{k,\text{MCMT}} = \underbrace{h_n^k a_n^k P_k}_{\text{useful signal power}} + \overbrace{h_n^k \sum_{\substack{i=1 \\ i \neq n}}^{N} a_i^k P_k}^{\text{intra-user interference}} + \mathcal{I}_{n,\text{MCMT}}^{k,\text{MBS}} + \mathcal{I}_{n,\text{MCMT}}^{k,\text{SBS}} + \sigma^2,
\tag{31}
$$

where $\mathcal{I}_{n,\text{MCMT}}^{k,\text{MBS}}$ and $\mathcal{I}_{n,\text{MCMT}}^{k,\text{SBS}}$ are the interferences at typical *k*-th tier *n*-th SU from MBSs and other SBSs, respectively.

*B. Outage Probability of NOMA-Based MCMT CN*

Based on Equation (30), the SINR at *m*-th MU after applying SIC in MCMT CN, denoted by $\Gamma_m^{\text{MCMT}}$, can be expressed as follows:

$$\Gamma_m^{\text{MCMT}} = \begin{cases} \dfrac{h_m a_m \rho}{h_m \rho \sum_{i=m+1}^{M} a_i + \bar{\rho} \mathcal{I}_{m,\text{MCMT}}^{\text{MBS}} + 1}, & \text{if } m\text{-th user is NCU} \\[4mm] \dfrac{h_m a_m \rho}{h_m \rho \sum_{i=m+1}^{M} a_i + \bar{\rho}\left(\mathcal{I}_{m,\text{MCMT}}^{\text{MBS}} + \mathcal{I}_{m,\text{MCMT}}^{\text{SBS}}\right) + 1}, & \text{if } m\text{-th user is CU.} \end{cases} \tag{32}$$

Now, similar to SCMT CN, at a particular time instant, MU can be NCU or CU (but not both simultaneously). As such, define the following two events.

$\varepsilon_5 = \{\text{Outage at tpical MU when it is NCU in MCMT CN}\}$
$\varepsilon_6 = \{\text{Outage at tpical MU when it is CU in MCMT CN}\}$

Based on the events $\varepsilon_1, \varepsilon_2, \varepsilon_5, \varepsilon_6$, the outage probability at the $m$th MU in MCMT CN, denoted by $P_m^{\text{MCMT}}$, can be expressed as follows:

$$P_m^{\text{MCMT}} = P_{\varepsilon_1} P_{\varepsilon_5} + P_{\varepsilon_2} P_{\varepsilon_6}. \tag{33}$$

Next, following similar steps to Equation (20), the probabilities $P_{\varepsilon_5}$ and $P_{\varepsilon_6}$ in Equation (33) can be expressed as follows:

$$P_{\varepsilon_5} = \int_0^{\infty} P_m^{\text{SCST}}\big|_{\tau_m^{\max} = \tilde{\tau}_m^{\max}} f_{\mathcal{I}_{m,\text{MCMT}}^{\text{MBS}}}(x)\,dx, \tag{34}$$

where $\tilde{\tau}_m^{\max} = \max\{\tilde{\tau}_1, ..., \tilde{\tau}_j, ...., \tilde{\tau}_m\}$, $\tilde{\tau}_j = \dfrac{\tilde{\varphi}_j\left(1 + \bar{\rho}\mathcal{I}_{m,\text{MCMT}}^{\text{MBS}}\right)}{\rho\left(a_j - \tilde{\varphi}_j \sum_{i=j+1}^{M} a_i\right)}$, $\tilde{\varphi}_j = 2^{R_j^{\text{MCMT}}} - 1$, $1 \leq j \leq m$, and $f_{\mathcal{I}_{m,\text{MCMT}}^{\text{MBS}}}(x)$ is the PDF of interference $\mathcal{I}_{m,\text{MCMT}}^{\text{MBS}}$.

Similarly, $P_{\varepsilon_6}$ can be written as:

$$P_{\varepsilon_6} = \int_0^{\infty}\int_0^{\infty} P_m^{\text{SCST}}\big|_{\tau_m^{\max} = \vartheta_m^{\max}} f_{\mathcal{I}_{m,\text{MCMT}}^{\text{MBS}}}(x)\, f_{\mathcal{I}_{m,\text{MCMT}}^{k,\text{SBS}}}(y)\,dxdy, \tag{35}$$

where $\vartheta_m^{\max} = \max\{\vartheta_1, ..., \vartheta_j, ...., \vartheta_m\}$, $\vartheta_j = \dfrac{\phi_j(1 + \bar{\rho}(x+y))}{\rho\left(a_j - \phi_j \sum_{i=j+1}^{N} a_i\right)}$, $\phi_j = 2^{R_j^{\text{MCMT}}} - 1$, $1 \leq j \leq n$, $R_j^{\text{MCMT}}$ is the targeted rate for $j$-th MU, $f_{\mathcal{I}_{m,\text{MCMT}}^{\text{MBS}}}(x)$ and $f_{\mathcal{I}_{m,\text{MCMT}}^{\text{SBS}}}(y)$ are the PDFs of interferences $\mathcal{I}_{m,\text{MCMT}}^{\text{MBS}}$ and $\mathcal{I}_{m,\text{MCMT}}^{\text{SBS}}$, respectively. Finally, using Equations (34) and (35) in Equation (33) derives the result for outage probability of $m$-th MU in NOMA-based MCMT CN.

Next, based on Equation (31), the SINR at $k$-th tier $n$-th SU after removing higher order users, denoted by $\Gamma_n^{k,\text{MCMT}}$ can be expressed as follows:

$$\Gamma_n^{k,\text{MCMT}} = \dfrac{h_n^k a_n^k \rho_k}{h_n^k \rho_k \sum_{i=n+1}^{N} a_i^k + \bar{\rho}\left(\mathcal{I}_{n,\text{MCMT}}^{k,\text{MBS}} + \mathcal{I}_{n,\text{MCMT}}^{k,\text{SBS}}\right) + 1}. \tag{36}$$

Now following similar approach in deriving Equation (20), the outage probability at the typical $k$-th tier $n$-th SU, denoted by $P_n^{k,\text{MCMT}}$ in MCMT CN, can be written as:

$$P_n^{k,\text{MCMT}} = \int_0^{\infty}\int_0^{\infty} P_m^{\text{SCST}}\big|_{\tau_m^{\max} = \bar{\vartheta}_{n,k}^{\max}} f_{\mathcal{I}_{n,\text{MCMT}}^{k,\text{MBS}}}(x)\, f_{\mathcal{I}_{n,\text{MCMT}}^{k,\text{SBS}}}(y)\,dxdy, \tag{37}$$

where $\bar{\vartheta}_{n,k}^{\max} = \max\{\bar{\vartheta}_{1,k}, ..., \bar{\vartheta}_{j,k}, ...., \bar{\vartheta}_{n,k}\}$, $\bar{\vartheta}_{j,k} = \dfrac{\bar{\phi}_{j,k}(1 + \bar{\rho}(x+y))}{\rho_k\left(a_j^k - \bar{\phi}_j^k \sum_{i=j+1}^{N} a_i^k\right)}$, $\bar{\phi}_j = 2^{R_j^{k,\text{MCMT}}} - 1$, $1 \leq j \leq n$, $R_j^{k,\text{MCMT}}$ is the targeted rate for $j$-th SU in tier $k$, $f_{\mathcal{I}_{n,\text{MCMT}}^{k,\text{MBS}}}(x)$ and $f_{\mathcal{I}_{n,\text{MCMT}}^{k,\text{SBS}}}(y)$ are the PDFs of

interference $\mathcal{I}_{n,\text{MCMT}}^{k,\text{MBS}}$ and $\mathcal{I}_{n,\text{MCMT}}^{k,\text{SBS}}$, respectively. This completes the discussion for outage probability in NOMA-based MCMT CNs.

*C. Sum-Rate Maximization Problem in NOMA-Based MCMT CN*

Let us consider an objective function $\mathcal{F}^{\text{MCMT}}$ which aims to maximize the sum-rate of the representative macro-cell. Similar to SCMT CN case, the objective function here can be defined as follows:

$$\mathcal{F}^{\text{MCMT}} = \sum_{m=1}^{M} \tilde{R}_m^{\text{MCMT}} + \sum_{k=2}^{K} \sum_{n=1}^{N} \tilde{R}_n^{k,\text{MCMT}}, \tag{38}$$

where $\tilde{R}_m^{\text{MCMT}}$ and $\tilde{R}_n^{k,\text{MCMT}}$ are the achievable rates of typical $m$-th MU and $k$-th tier $n$-th SU in MCMT CN. Now, based on Equation (38), the optimization problem can be formulated as follows:

$$\max \mathcal{F}^{\text{MCMT}} \tag{39}$$

subject to:

$\mathbf{C_1}:\quad R_m^{\text{MCMT}} \leq \tilde{R}_m^{\text{MCMT}}$

$\mathbf{C_2}:\quad R_n^{k,\text{MCMT}} \leq \tilde{R}_n^{k,\text{MCMT}}$

$\mathbf{C_3}:\quad \displaystyle\sum_{m=1}^{M} a_m P \leq P$

$\mathbf{C_4}:\quad \displaystyle\sum_{n=1}^{N} a_n^k P_k \leq P_k, \quad 2 \leq k \leq K$

$\mathbf{C_5}:\quad \phi_j \displaystyle\sum_{i=j+1}^{M} a_i \leq a_j, \quad 1 \leq j \leq M$

$\mathbf{C_6}:\quad \bar{\phi}_j^k \displaystyle\sum_{i=j+1}^{N} a_i^k \leq a_j^k, \quad 1 \leq j \leq N$

$\mathbf{C_7}:\quad \mathcal{I}_{m,\text{MCMT}}^{\text{MBS}} + \mathcal{I}_{m,\text{MCMT}}^{\text{SBS}} < h_m a_m P, \quad 1 \leq m \leq M$

$\mathbf{C_8}:\quad \mathcal{I}_{n,\text{MCMT}}^{k,\text{MBS}} + \mathcal{I}_{n,\text{MCMT}}^{k,\text{SBS}} < h_n^k a_n^k P_k, \quad 1 \leq n \leq M, 2 \leq k \leq K,$

The constraints in optimization problem Equation (39) can be explained in a similar way as was described in optimization problem Equation (24) for SCMT CN.

*3.2. NOMA Applied to Device-To-Device Communications*

In current literature, the application of NOMA to device-to-device (D2D) communication is relatively less explored. Particularly, the existing works pay little attention in investigating D2D networks where D2D nodes are communicating with the aid of NOMA transmission scheme. In [158], the authors considered a cooperative NOMA system with full-duplex (FD) functionality. The BS applies NOMA protocol to serve strong and weak users. In order to enhance the outage performance of the weak NOMA user, the authors proposed to utilize FD at the strong user. Under this setting, the users communicate with each other by utilizing OMA based cooperative D2D communication. Furthermore, the authors in [159] considered a cellular-network with underlay D2D communications, where D2D users can also operate in FD mode. In order to select between D2D and FD modes, the authors proposed a novel selection criteria. Based on stochastic geometry techniques, the performance of considered network is characterized by deriving closed-form outage probability expressions. The spatial locations of D2D users are modeled by the Poisson point process (PPP), which may not be an accurate distribution for modeling D2D locations because it cannot capture the key characteristics

of device clustering and spatial separation for D2D communications [160]. A MIMO-NOMA-based multiuser downlink cellular network is considered in [161]. Moreover, D2D communication in underlay mode is also invoked in order to further enhance the spectral efficiency of the considered network. As such, this configuration constitutes intra-beam interference as well as interference at D2D users due to BS transmission. Consequently, in order to mitigate both types of interference, the authors proposed two beamforming methods. Further, the performance of both cellular and D2D users is jointly investigated by formulating an optimization problem. However, a conventional OMA-based paired D2D communication is considered and the system model is limited to single-cell scenario only. The authors in [162] considered a paired D2D communication underlying a NOMA-based cellular network in which a single hybrid access point is used to receive from both D2D and cellular users. In particular, they investigate the problem of resource allocation and propose a low complexity energy efficient algorithm for the D2D pair while meeting the QoS requirements of cellular users. The results demonstrated fast convergence of their proposed algorithm to an optimal solution while achieving superior energy efficiency over the existing schemes. A major limitation of their network model is that they considered only a single D2D active pair, whereas in real situations, there can be more than one active D2D pairs in the network, which requires joint optimization of energy efficiency of all active D2D pairs. Similarly, the problem of resource allocation in terms of power control and channel assignment for D2D communication underlying a NOMA-based cellular network is investigated in [163]. The authors derived the optimal conditions for cellular users under which power control on each subchannel is conducted. Based on the derived conditions, they proposed an algorithm to maximize the sum rate of D2D pairs while simultaneously satisfy the QoS requirements of cellular users. However, the considered cellular network is restricted to the case of a single cell, thereby neglecting the impact of interference from multiple BSs. In addition, NOMA is not utilized for D2D communications, which may result in less efficient resource utilization. The concept of NOMA assisted D2D relaying is proposed in [164], where the transmission is composed of two phases. In the first phase, the BS transmits NOMA signal to near and far users. By exploiting the overhearing of this NOMA transmission at intermediate D2D relay node, in the second phase, this relay node transmits superimposed overheard NOMA signal from first phase and its own signal for the D2D receiver using NOMA. This helps to boost the performance of far user. Although NOMA is used by the D2D relay node, it is still communicating with a single D2D receiver to form a paired D2D communication.

The work of [165] can be regarded as a pioneer in investigating NOMA-based D2D communications, where authors came up with a notion of the D2D group. The key idea of group D2D communication is that a D2D transmitter (DT) acts like a BS and applies the NOMA transmission technique in order to communicate with multiple D2D receivers (DRs). The introduction of group D2D communication would generate a network interference. As such, in order to realize a group D2D network, an optimal resource allocation algorithm is proposed in order to efficiently perform interference management. Furthermore, the work in [166] can be regarded as the extension of [165] in which a joint subchannel and power allocation algorithm is proposed in order to maximize the sum-rate of D2D users. The results demonstrate that the proposed algorithm is able to achieve near optimal performance. However, again the system model is limited to a single-cell scenario and proper interference modeling and characterization would be required for multi-cell extension. The authors in [167] considered a large-scale D2D network and proposed a cooperative hybrid automatic repeat request (HARQ) scheme based on NOMA. The performance of D2D users in the considered system is evaluated in terms of outage probability and throughput. The results reveal that the proposed cooperative HARQ assisted NOMA scheme outperforms both non-cooperative and OMA based networks. However, the authors considered only two user NOMA in this work and also assumed that one NOMA user always exist in close proximity of the D2D source. This would be a rather strong assumption which may not generally hold. A QoS-based NOMA group D2D communication system is proposed in [168]. The authors introduced a notion of QoS based power allocation. Based on stochastic geometry techniques, the closed-form analytical expressions for outage probability are

derived. The simulation results demonstrate that the proposed QoS based NOMA group D2D system achieves lower outage probability than the conventional OMA-based paired D2D communication. The performance of cooperative D2D network with NOMA is investigated in [169] where each D2D link exhibits independent Nakagami-m fading. In order to evaluate the system performance, closed-form expressions for outage probability and throughput are presented. In addition, numerical results reveal the accuracy of the derived analytical results. A cooperative D2D system is considered in [170] where BS communicates with all the users simultaneously by applying the NOMA protocol. The authors proposed two decoding schemes in order to study the impact of weak channel and different decoding strategies on the performance of the considered network. Further, the authors presented the expressions for ergodic sum-rate, outage probability, and outage capacity to analyze the system performance. Numerical results are also presented to highlight the superiority of the proposed schemes over conventional NOMA and OMA. The authors in [171] considered a NOMA system with underlay D2D communication. In order to avoid the impact of interference on the performance of cellular user, the authors have proposed a strategy for D2D users by which they can select to operate between interlay and underlay modes. Further, a joint D2D mode selection and resource allocation optimization problem is formulated which aims to maximize the sum-rate of the considered system. Simulation results reveal that utilizing the proposed mode selection scheme for D2D in the considered network significantly enhances the overall system sum-rate and D2D access rate. The authors in [172] consider a D2D enabled NOMA network and proposed a novel framework in order to maximize the performance of the D2D communications in terms of energy efficiency and sum-rate. As such, the optimization problem is formulated which jointly considers resource block assignment and power allocation based on SIC decoding order of cellular users. The results show that the considered network under the proposed optimization algorithm achieves superior performance compared to the conventional OMA system. The problem of resource allocation for NOMA-based D2D network with underlay CN is investigated in [173]. An optimization problem is formulated in order to optimize the total transmit power for all the users by jointly considering subchannel, user pairing, and power control. The authors reported that the minimization of total transmit power for the network has a high dependency on subchannel assignment and user pairing. The problem of joint D2D group association and channel assignment is investigated in [174] for uplink multi-cell NOMA systems. In order to utilize cellular channels at each BS by D2D groups, the authors proposed SINR maximization based multi-objective power allocation solution with constraint of meeting QoS requirements. In addition, the problems of D2D group association and channel assignment are solved by applying principles of matching theory. Simulation results reveal the effectiveness of proposed algorithms by achieving comparable SINR for each user and D2D receiver to the joint group association, channel, and power allocations. The issue of security in the presence of eavesdropper for D2D enabled NOMA-based CNs are investigated in [175]. In order to resolve the security problem in the considered network, the authors have formulated and solved a joint power optimization problem which targets to maximize the secrecy sum-rate of the entire network. In addition, closed form expressions for connection outage and secrecy outage probabilities are derived to characterize the performance. Simulation results are provided to validate the accuracy of the derived analytical results as well as the superior performance of proposed algorithms in enhancing joint security of the network. The authors in [176] considered an uplink D2D communication based on NOMA and investigated the problem of joint sub-channel and power allocation in order to maximize the overall energy efficiency and throughput of the considered system. Simulation results reveal that under the proposed optimal power allocation algorithm, the considered system achieves superior performance than the existing schemes in terms of energy efficiency and throughput. In order to enhance the data rate of D2D communications, the authors in [177] considered a D2D and hybrid CN, where joint user pairing and power control optimization problem is formulated while taking into account the decoding threshold for cellular users. As such, an optimal power control algorithm is proposed to solve the optimization problem. The results show that the proposed algorithm achieves superior data rate when compared to the current techniques.

### 3.2.1. NOMA-Based Group D2D Communications

A general group D2D (GD2D) network based on NOMA protocol is shown in Figure 9. It can be observed that D2D users are distributed randomly over the entire two-dimensional plane. Further, it is assumed that the D2D network is operating in an inband mode with underlay MCST CN. In addition, consider that DT, referred as group transmitter (GT), is communicating with multiple DRs via NOMA protocol. Note that at particular time instant, the selection of GT is performed by the BS. For analysis purposes, we consider a representative macro-cell in which $M$ MUs and multiple GTs are randomly distributed, as shown in Figure 9. Further, let $\mathcal{R}_\mathcal{D}$ and $\mathcal{R}_{\text{GT}}$ model the coverage of MBS and a GT, respectively. In addition, for ease of exposition, it is considered that each GT is serving $L$ DRs via NOMA protocol, which are assumed to be randomly distributed inside its coverage. Without loss of generality, assume that channel gains of MUs and DRs (in coverage of representative GT) are ordered as $h_1 \le ... \le h_M$ and $\tilde{h}_1 \le ... \le \tilde{h}_L$, respectively, where $\tilde{h}_l = \frac{\tilde{g}_l}{\tilde{d}_l^\alpha}$. Here, $\tilde{g}_l$ and $\tilde{d}_l$ represent the fading (power) gain and distance between $l$-th DR and representative GT, respectively, and $1 \le l \le L$. Consequently, the power allocation coefficients for MUs and DRs are sorted as, $a_1 \ge ... \ge a_M$ and $\tilde{a}_1 \ge ... \ge \tilde{a}_L$, respectively. The received power at the $m$-th MU and $l$-th DR, denoted by $P_m^{\text{GD2D}}$ and $P_l^{\text{GD2D}}$, can be expressed as follows:

$$P_m^{\text{GD2D}} = \underbrace{h_m a_m P}_{\text{useful signal power}} + \overbrace{h_m \sum_{\substack{i=1 \\ i \ne m}}^{M} a_i P}^{\text{intra-user interference}} + \mathcal{I}_{m,\text{GD2D}}^{\text{MBS}} + \mathcal{I}_{m,\text{GD2D}}^{\text{GT}} + \sigma^2, \tag{40}$$

$$P_l^{\text{GD2D}} = \underbrace{\tilde{h}_l \tilde{a}_l P_{\text{GT}}}_{\text{useful signal power}} + \overbrace{\tilde{h}_l \sum_{\substack{i=1 \\ i \ne l}}^{L} \tilde{a}_i P_{\text{GT}}}^{\text{intra-user interference}} + \mathcal{I}_{l,\text{GD2D}}^{\text{MBS}} + \mathcal{I}_{l,\text{GD2D}}^{\text{GT}} + \sigma^2, \tag{41}$$

where $P_{\text{GT}}$ is the transmission power of GT, $\mathcal{I}_{m,\text{GD2D}}^{\text{MBS}}$, $\mathcal{I}_{m,\text{GD2D}}^{\text{GT}}$ represent the interference at $m$-th MU from other MBSs and GTs, respectively, and $\mathcal{I}_{l,\text{GD2D}}^{\text{MBS}}$, $\mathcal{I}_{l,\text{GD2D}}^{\text{GT}}$ denote the interferences at $l$-th DR from MBSs and other GTs, respectively.

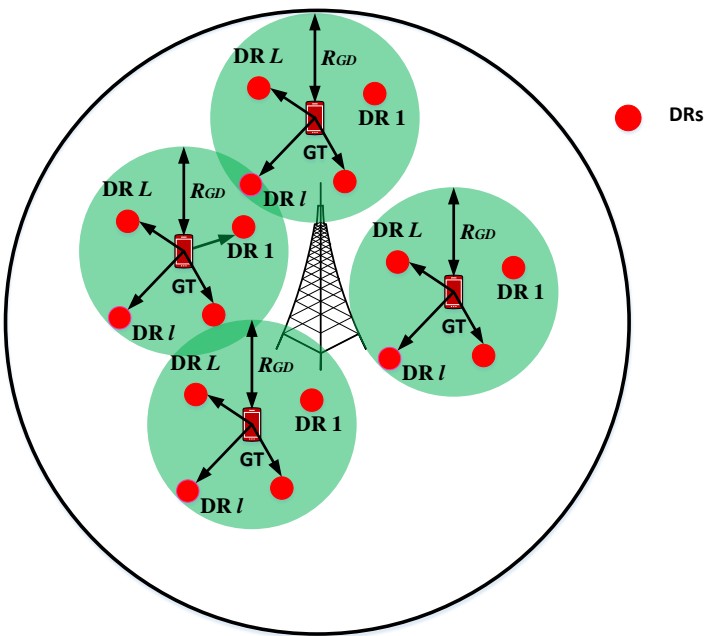

**Figure 9.** Group device-to-device (D2D) network with underlay cellular network.

### 3.2.2. Outage in NOMA-Based Group D2D Network

This subsection describes the outage probability at $m$-th MU and $l$-th DR in a representative macro-cell and *GT* coverage, respectively. Based on Equation (40), the SINR after applying SIC at $m$-th MU in NOMA-based GD2D network, denoted as $\Gamma_m^{\text{GD2D}}$, can be written as follows:

$$\Gamma_m^{\text{GD2D}} = \frac{h_m a_m \rho}{h_m \rho \sum_{\substack{i=1 \\ i \neq m}}^{M} a_i + \bar{\rho}\left(\mathcal{I}_{m,\text{GD2D}}^{\text{MBS}} + \mathcal{I}_{m,\text{GD2D}}^{\text{GT}}\right) + 1}. \tag{42}$$

Next, based on Equation (42) and following similar steps to deriving Equation (35), the outage probability at $m$-th MU in NOMA-based GD2D underlay CN, denoted as $P_m^{\text{GD2D}}$, can be given as:

$$P_m^{\text{GD2D}} = \int_0^\infty \int_0^\infty P_m^{\text{SCST}}\big|_{\tau_m^{\max} = \tilde{\vartheta}_m^{\max}} f_{\mathcal{I}_{m,\text{GD2D}}^{\text{MBS}}}(x) \, f_{\mathcal{I}_{m,\text{GD2D}}^{\text{SBS}}}(y) \, dx dy, \tag{43}$$

where $\tilde{\vartheta}_m^{\max} = \max\left\{\tilde{\vartheta}_1, ..., \tilde{\vartheta}_j, ...., \tilde{\vartheta}_l\right\}$, $\tilde{\vartheta}_j = \frac{\tilde{\phi}_j(1+\bar{\rho}(x+y))}{\rho\left(a_j - \tilde{\phi}_j \sum_{i=j+1}^{N} a_i\right)}$, $\tilde{\phi}_j = 2^{R_j^{\text{GD2D}}} - 1$, $1 \leq j \leq l$, $R_j^{\text{GD2D}}$ is the targeted rate for $j$-th MU, $f_{\mathcal{I}_{m,\text{GD2D}}^{\text{MBS}}}(x)$ and $f_{\mathcal{I}_{m,\text{GD2D}}^{\text{SBS}}}(y)$ are the PDFs of interferences $\mathcal{I}_{m,\text{GD2D}}^{\text{MBS}}$ and $\mathcal{I}_{m,\text{GD2D}}^{\text{SBS}}$, respectively.

Similarly, based on Equation (41), SINR after applying SIC at $l$-th DR in NOMA-based GD2D network, denoted as $\Gamma_l^{\text{GD2D}}$, can be expressed as follows:

$$\Gamma_l^{\text{GD2D}} = \frac{\tilde{h}_l \tilde{a}_l \rho_l}{\tilde{h}_l \rho_l \sum_{i=l+1}^{L} \tilde{a}_l + \bar{\rho}\left(\mathcal{I}_{l,\text{GD2D}}^{\text{MBS}} + \mathcal{I}_{l,\text{GD2D}}^{\text{SBS}}\right) + 1}, \tag{44}$$

where $\rho_l = \frac{P_{\text{GT}}}{\sigma^2}$. Now based on Equation (44), the outage probability at $l$-th DR, denoted as $P_l^{\text{GD2D}}$, can be expressed as:

$$P_l^{\text{GD2D}} = \int_0^\infty \int_0^\infty P_m^{\text{SCST}}\big|_{\tau_m^{\max}=\hat{\vartheta}_l^{\max}} f_{\mathcal{I}_{l,\text{GD2D}}^{\text{MBS}}}(x)\, f_{\mathcal{I}_{l,\text{GD2D}}^{\text{SBS}}}(y)\, dx dy, \tag{45}$$

where $\hat{\vartheta}_l^{\max} = \max\left\{\hat{\vartheta}_1, ..., \hat{\vartheta}_j, ...., \hat{\vartheta}_l\right\}$, $\hat{\vartheta}_j = \dfrac{\hat{\phi}_j(1+\bar{\rho}(x+y))}{\rho\left(a_j - \hat{\phi}_j \sum_{i=j+1}^N a_i\right)}$, $\hat{\phi}_j = 2^{R_j^{\text{GD2D}}} - 1$, $1 \le j \le l$, $R_j^{\text{GD2D}}$ is the targeted rate for *j*-th DR, $f_{\mathcal{I}_{l,\text{GD2D}}^{\text{MBS}}}(x)$ and $f_{\mathcal{I}_{l,\text{GD2D}}^{\text{SBS}}}(y)$ are the PDFs of interferences $\mathcal{I}_{l,\text{GD2D}}^{\text{MBS}}$ and $\mathcal{I}_{l,\text{GD2D}}^{\text{SBS}}$, respectively. This completes the discussion for outage probability in NOMA-based GD2D networks.

### 3.2.3. Sum-Rate Maximization Problem in NOMA-Based GD2D Networks

Similar to CN scenario, let us denote by $\mathcal{F}^{\text{GD2D}}$, as the objective function in order to maximize the overall sum-rate of the NOMA-based GD2D network. The function $\mathcal{F}^{\text{GD2D}}$ can be defined as follows:

$$\begin{aligned} \mathcal{F}^{\text{GD2D}} &= \sum_{m=1}^M \tilde{R}_m^{\text{GD2D}} + \sum_{l=1}^L \tilde{R}_l^{\text{GD2D}} \\ &= \sum_{m=1}^M \log_2\left(1 + \Gamma_m^{\text{GD2D}}\right) + \sum_{l=1}^L \log_2\left(1 + \Gamma_l^{\text{GD2D}}\right), \end{aligned} \tag{46}$$

where $\tilde{R}_m^{\text{GD2D}}$ and $\tilde{R}_l^{\text{GD2D}}$ represent the achievable rates for *m*-th MU and *l*-th DR, respectively. Based on definition of $\mathcal{F}^{\text{GD2D}}$, the optimization problem can now be formulated as follows:

$$\max \mathcal{F}^{\text{GD2D}} \tag{47}$$

subject to:

$$\mathbf{C_1}: \quad R_m^{\text{GD2D}} \le \tilde{R}_m^{\text{GD2D}}$$

$$\mathbf{C_2}: \quad R_n^{k,\text{MCMT}} \le \tilde{R}_n^{k,\text{SCMT}}$$

$$\mathbf{C_3}: \quad \sum_{m=1}^M a_m P \le P$$

$$\mathbf{C_4}: \quad \sum_{l=1}^L a_l P_l \le P_l,$$

$$\mathbf{C_5}: \quad \tilde{\phi}_j \sum_{i=j+1}^M a_i \le a_j, \quad 1 \le j \le M$$

$$\mathbf{C_6}: \quad \hat{\phi}_j \sum_{i=j+1}^L a_i \le a_j, \quad 1 \le j \le L$$

$$\mathbf{C_7}: \quad \mathcal{I}_{m,\text{GD2D}}^{\text{MBS}} + \mathcal{I}_{m,\text{GD2D}}^{\text{GT}} < h_m a_m P, \quad 1 \le m \le M$$

$$\mathbf{C_8}: \quad \mathcal{I}_{l,\text{GD2D}}^{\text{MBS}} + \mathcal{I}_{l,\text{GD2D}}^{\text{GT}} < \tilde{h}_l a_l P_l, \quad 1 \le l \le L.$$

The constraints $\mathbf{C_1}$–$\mathbf{C_8}$ in Equation (47) follow similar explanation as the constraints described in optimization Equation (39) for MCMT CN.

### 3.3. NOMA Applied to Wireless Sensor Networks

The current literature is scarce in investigating the application of NOMA to WSNs. The work in [178] can be regarded as pioneer in investigating the NOMA-based WSNs. By utilizing stochastic geometry tools, the authors proposed and modeled a NOMA-based ubiquitous WSN in which sensors and sink nodes are considered to be randomly distributed over the entire two-dimensional plane. Further, the cross-technology (CT) nodes operating in the same frequency band are also considered to be co-located with the WSN. In order to evaluate the performance of the considered network,

the authors derived closed-form expressions for outage probability, diversity, and throughput. In addition, computational complexity analysis is also presented in order to justify the proposal of NOMA for WSN. The simulation results demonstrate the superior performance of NOMA-based WSN over conventional WSN utilizing OMA for transmission. The authors in [179] considered a two-hop NOMA-based WSN with energy harvesting. Based on two types of relying, the authors proposed a novel energy harvesting algorithm. In order to evaluate the performance, closed-form expressions for outage probability and ergodic rate are derived. Numerical results demonstrate that the proposed scheme is able to achieve significantly lower outage probability and superior rate compared to the existing techniques. Furthermore, NOMA-based WSN is considered in [180] with time switching energy harvesting technique. Closed-form analytical expressions for outage probability and achievable rate are derived. The Monte Carlo simulations are performed in order to validate the accuracy of the derived results. The application of NOMA-based WSN for smart agriculture system is investigated in [181]. The authors considered a relay-aided uplink NOMA system and characterize the performance by deriving closed-form analytical expressions for outage probability and average sum-rate. The simulation results demonstrate that the proposed relay-aided NOMA-based scheme achieves superior performance over the conventional OMA technique for WSN in agriculture. The authors in [182] considered a WSN based on NOMA, where time-switching and power-splitting based relaying protocols are proposed in order to deploy energy harvesting. Further, closed-form expression for outage probability and throughput are derived to characterize the network performance. In addition, an optimization problem is formulated for maximizing the data rate of the entire system. Simulation results reveal that the performance of considered improves significantly under the two proposed algorithms. In [183], the authors considered NOMA-based frequency hopping ad hoc network, where different users are grouped together to form NOMA clusters. The authors applied stochastic geometry tools to analyze the performance of considered network and derived closed-form expressions for coverage probability and average sum-rate. Numerical results demonstrate the accuracy of the derived analytical expressions. In addition, the results also reveal that the by choosing appropriate cluster radius, number of frequency points, user power allocation, and cluster density could enhance the coverage probability of the considered network.

### 3.3.1. NOMA-Based WSNs

Consider a WSN as shown in Figure 10, where sensors and sink nodes are randomly distributed across the entire two-dimensional plane. Since, WSNs generally operate in license a free band, therefore, there are other cross-technology (CT) nodes co-exist in the considered WSN, as depicted in the Figure 10. Further, assume that each sink node is located at the center of a disc with radius $\mathcal{R}_{\mathrm{SINK}}$, modeling the coverage of the sink node. Moreover, the sink node is communicating with multiple sensor nodes by applying NOMA transmissions. In addition, let there are total of $U$ sensor nodes distributed randomly inside the coverage of each sink node. Furthermore, for analysis purposes, we consider a representative sink node. Without loss of generality, the sensor nodes are ordered as $\hat{h}_1 \leq ... \leq \hat{h}_U$, where $\hat{h}_u = \frac{\hat{g}_u}{\hat{d}_u^\alpha}$, $1 \leq u \leq U$. Here, $\hat{g}_u$ and $\hat{d}_u$ are the fading (power) and distance between the $u$-th sensor and representative sink node, respectively. Consequently, under this network arrangement, there would be an interference at typical $u$-th sensor node in the coverage of representative sink node from NOMA transmissions of other sink and CT nodes. Note that typically CNs operate in a licensed spectrum and hence there would be no interference from CN to the sensors and sinks in a WSN. As a result, the received power at the $u$-th sensor, denoted by $P_u^{\mathrm{WSN}}$, can be written as follows:

$$P_m^{\mathrm{GD2D}} = \underbrace{\hat{h}_u a_u P_{\mathrm{SINK}}}_{\text{useful signal power}} + \overbrace{\hat{h}_u \sum_{\substack{i=1 \\ i \neq u}}^{U} a_i P_{\mathrm{SINK}}}^{\text{intra-user interference}} + \mathcal{I}_{u,\mathrm{WSN}}^{\mathrm{SINK}} + \mathcal{I}_{u,\mathrm{WSN}}^{\mathrm{CT}} + \sigma^2, \qquad (48)$$

where $P_{\text{SINK}}$ is the transmission power of sink node, $\mathcal{I}_{u,\text{WSN}}^{\text{SINK}}$ and $\mathcal{I}_{u,\text{WSN}}^{\text{CT}}$ represent the interference at the $u$-th sensor node from other sink nodes and CT nodes, respectively.

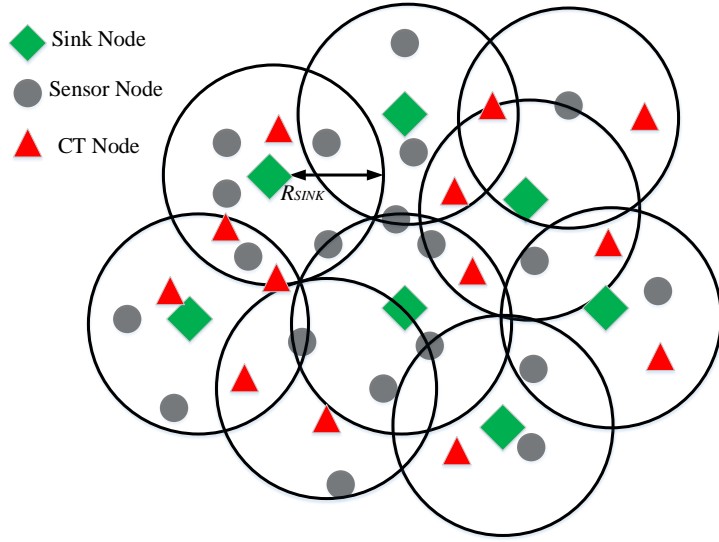

**Figure 10.** NOMA-based wireless sensor network (WSN).

3.3.2. Outage in NOMA-Based WSNs

This subsection expresses the outage probability at the $u$-th sensor node in the considered NOMA-based WSN. After applying SIC at the $u$-th sensor node, the SINR denoted as $\Gamma_u^{\text{WSN}}$, can be written using Equation (48) as follows:

$$\Gamma_u^{\text{WSN}} = \frac{\hat{h}_u a_u \rho_u}{\hat{h}_u \rho_u \sum_{i=u+1}^{U} a_i + \bar{\rho}\left(\mathcal{I}_{u,\text{WSN}}^{\text{SINK}} + \mathcal{I}_{u,\text{WSN}}^{\text{CT}}\right) + 1}, \tag{49}$$

where $\rho_u = \frac{P_{\text{SINK}}}{\sigma^2}$. Next, following similar procedure to deriving Equation (20), the outage probability at $u$-th sensor node, denoted by $\text{P}_u^{\text{WSN}}$, can be expressed as follows:

$$\text{P}_u^{\text{WSN}} = \int_0^\infty \int_0^\infty \text{P}_m^{\text{SCST}}\big|_{\tau_m^{\max} = \breve{\vartheta}_u^{\max}} f_{\mathcal{I}_{u,\text{WSN}}^{\text{SINK}}}(x)\, f_{\mathcal{I}_{u,\text{WSN}}^{\text{CT}}}(y)\, dx\, dy, \tag{50}$$

where $\breve{\vartheta}_u^{\max} = \max\left\{\breve{\vartheta}_1, ..., \breve{\vartheta}_j, ...., \breve{\vartheta}_u\right\}$, $\breve{\vartheta}_j = \frac{\breve{\phi}_j(1+\bar{\rho}(x+y))}{\rho\left(a_j - \breve{\phi}_j \sum_{i=j+1}^{U} a_i\right)}$, $\breve{\phi}_j = 2^{R_j^{\text{WSN}}} - 1$, $1 \leq j \leq u$, $R_j^{\text{WSN}}$ is the targeted rate for $j$-th SN, $f_{\mathcal{I}_{u,\text{WSN}}^{\text{SINK}}}(x)$ and $f_{\mathcal{I}_{u,\text{WSN}}^{\text{CT}}}(y)$ are the PDFs of interferences $\mathcal{I}_{l,\text{WSN}}^{\text{SINK}}$ and $\mathcal{I}_{l,\text{WSN}}^{\text{CT}}$, respectively. This completes the discussion for outage probability in NOMA-based WSNs.

3.3.3. Sum-Rate Maximization Problem in NOMA-Based WSNs

In order to maximize the sum-rate in NOMA-based WSN, let us consider an objective function, denoted by $\mathcal{F}^{\text{WSN}}$, and defined as:

$$\begin{aligned} \mathcal{F}^{\text{WSN}} &= \sum_{u=1}^{U} \tilde{R}_u^{\text{WSN}} \\ &= \log_2\left(1 + \Gamma_u^{\text{WSN}}\right), \end{aligned} \tag{51}$$

where $\tilde{R}_u^{\mathrm{WSN}}$ is the achievable rate of $u$-th sensor node. Now, based on $\mathcal{F}^{\mathrm{WSN}}$ in Equation (51), the optimization problem can be formulated as follows:

$$\max \mathcal{F}^{\mathrm{WSN}} \tag{52}$$

subject to:

$$\mathbf{C_1}: \quad R_u^{\mathrm{WSN}} \leq \tilde{R}_u^{\mathrm{WSN}}$$

$$\mathbf{C_2}: \quad \sum_{u=1}^{U} a_u P_u \leq P_u$$

$$\mathbf{C_3}: \quad \check{\varphi}_j \sum_{i=j+1}^{U} a_i \leq a_j, \quad 1 \leq j \leq U$$

$$\mathbf{C_4}: \quad \mathcal{I}_{u,\mathrm{WSN}}^{\mathrm{SINK}} + \mathcal{I}_{u,\mathrm{WSN}}^{\mathrm{CT}} < \hat{h}_u a_u P_u, \quad 1 \leq u \leq U.$$

Note that the constraints in problem Equation (52) can be explained in a similar manner as were described in Equation (29) for MCST CN.

## 4. Research Challenges and Future Trends

The prior arts on NOMA have extensively investigated many NOMA specific issues and its integration with several communication technologies. However, there are still multiple unresolved issues which need to be addressed in order to further enhance the performance of NOMA systems. As such, this section discusses some key design challenges for NOMA and indicate potential research directions to solve theses issues.

### 4.1. Alternate Receiver for NOMA

In current literature, NOMA has been extensively investigated with the SIC receiver. While carrying out theoretical analysis of NOMA systems with SIC, it has been very commonly assumed that all the higher order users are perfectly removed at say $m$-th user receiver i.e., perfect cancellation is assumed. However, this assumption does not hold generally, rather in practical situations the cancellation is mostly imperfect (there is some residual interference left). Therefore, there is a need to consider this imperfect cancellation aspect in theoretical analysis of NOMA-based wireless systems. In addition, the performance of SIC technique is sensitive to power levels among users. This implies that appropriate rate and power allocation NOMA with SIC would be required and failing to comply with this condition would always result in complete outage [70]. Furthermore, there is a problem of error propagation in SIC. This means that whenever any of the higher-order user has been decoded incorrectly, the error will sequentially propagate to decoding of subsequent lower-order users [184]. As a consequence, these limitations of SIC receiver limit the NOMA functionality. Therefore, in order to resolve the aforementioned SIC related issues, either existing SIC design could be improved or an alternate receiver for NOMA can be considered. The alternate receiver design here means to implement a receiver which does not utilize successive cancellation procedure for multiuser detection. One promising approach is to apply machine learning techniques in order to obtain predictions of NOMA users and then utilize these estimates to aid SIC decoding. Another potential solution is to consider parallel interference cancellation (PIC) technique, message passing algorithm [25], and deep learning methods to design an alternate receiver for NOMA.

### 4.2. Hybrid Multiple Access

NOMA systems apply SC and are hence inherently interference limited due to the presence of intra-user interference. As such, the performance gain of NOMA over conventional OMA is limited in the lower SNR regime [70]. Consequently, the NOMA scheme in its pure form appears to be not very suitable for wireless systems that are operating at low SNRs (IoT, D2D, M2M communications,

and so on). In order to resolve this problem, a hybrid MA based on combination of NOMA and OMA could be designed, which is capable of switching between NOMA and OMA operation modes under conditions of optimizing overall system throughput.

### 4.3. Consideration of Imperfect CSI

The existing literature on NOMA commonly assumes a perfect CSI order to either perform resource allocation at BS or suppress multiuser interference at the user receiver. However, this assumption is very strong as perfect CSI is not possible to obtain in practical NOMA systems. Consequently, real-time NOMA networks operate under channel estimation errors. As such, it is required to consider imperfect CSI and channel estimation errors in theoretical analysis of NOMA systems. Some attempts are made in existing literature in order to account the impact of channel estimation errors in theoretical analysis [185–188]. Nevertheless, the upcoming 5G systems are envisioned to support a massive number of users, which in turn could increase inter-user interference, resulting in enhanced channel estimation errors. Therefore, more sophisticated technique and algorithms are required in order to obtain accurate channel estimation in practical NOMA systems.

### 4.4. NOMA-Based Data Offloading in 5G Networks

The upcoming 5G wireless systems are anticipated to operate in a hyper-connected environment, where a massive number of connected devices will generate enormous amount of data on the backbone network. This poses a great challenge for network operators in order to store, process, and transmit this big data. A typical example of such a scenario could be a congested macro-cell. One promising solution to solve this problem is to offload data. For example, a MBS in a congested macro-cell can offload data to SBSs or to Wi-Fi networks. However, utilizing conventional OMA based offloading would result in increased latency, particularly under scenarios of huge data size and sources. Consequently, this situation may not be feasible particularly for delay-sensitive applications. As such, by virtue of SC, NOMA has a potential to efficiently offload huge amounts of data and users simultaneously. Therefore, it is required to design advance-offloading algorithms based on NOMA to resolve the problem of tackling big data in upcoming 5G wireless networks.

### 4.5. NOMA-Aided Full Duplex Networks

The FD duplex technique has a potential to enhance the spectral efficiency by simultaneously transmitting and receiving over the same frequency channel. In order to meet the ambitious demands of 5G wireless systems, the integration of NOMA with FD communication could be a promising solution to enhance overall system capacity. In current literature, some attempts are made to analyze NOMA-based FD networks, see [189–193] for quick reference. However, FD and NOMA systems are limited by self-interference and intra-user interference, respectively, which may degrade the system performance. Consequently, an efficient low-complexity receiver design would be required to realize NOMA-aided FD networks. Furthermore, advanced resource and power allocation algorithms would be required which could result in minimizing self-interference and intra-user interference and maximizing the overall sum-rate of the system. Therefore, investigating NOMA-aided FD networks and its related issues could be a promising future research direction.

## 5. Conclusions

This survey has described the fundamental concept and potential benefits of NOMA protocol, which has been admitted as a new member of MA schemes. More precisely, the application of NOMA to CNs, D2D communications, and WSNs has been reviewed and discussed in detail. Moreover, outage probability expressions are presented and sum-rate maximization problems are formulated for each NOMA-based wireless network. The presented analytical expressions for a given NOMA-aided system (CN, D2D, WSN) can be easily tailored based on the specific assumptions related to the spatial topology of the network. Furthermore, this study also highlighted some of the key NOMA-related design issues

along with some potential future research directions. It is strongly believed and anticipated that NOMA will be a key player in upcoming 5G networks with exceptional potential of supporting massive connectivity and low latency.

**Author Contributions:** All authors contributed equally towards planning, writing, and reviewing of this work.

**Funding:** This research received no external funding.

**Conflicts of Interest:** The authors declare no conflict of interest.

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
