# Peer review of "A Survey on Application of Non-Orthogonal Multiple Access to Different Wireless Networks"

_electronics, doi:10.3390/electronics8111355_

Round 1

Reviewer 1 Report

This survey had described the fundamental concept and potential benefits of NOMA protocol, which has been admitted as a new member of MA schemes. More precisely, the application of NOMA to CNs, D2D communications, and WSNs has been reviewed and discussed in detail. Moreover, outage probability expressions are presented and sum-rate maximization problems are formulated for each NOMA based wireless network. Furthermore, this study has also highlighted some of the key NOMA related design issues along with some potential future research directions.

Overall, the paper presents a reasonable scientific contribution. However, the reviewer has few comments given below.

It would be a great idea if the authors could add taxonomy for section 3 “Application of NOMA to Wireless Networks”.

As this is a survey paper, the mathematical contents should be kept as low as possible.

Authors must review the acronyms and their extension (on the first appearance). Apart from abstract, the full name of the abbreviations should be provided for the first time appearance, and then only the abbreviations should be used. “Please provide the full form of “ATSC”, “NTT DOCOMO”, “MUD”, etc. at its first appearance.

Please proofread the paper again to correct the typographical errors. The authors should carefully revise this paper as there are some spellings, and grammatical errors.

Also, there is lot of room for language improvement.

To generate more research interest, the authors should include some future works in the article.

Please replace some older references with the related and latest ones.

The suggested revisions must be incorporated in the revised manuscript.

Reviewer 2 Report

This survey paper on NOMA has reviewed many papers in NOMA and listed some challenges for future research. The reviewer believes the motivation for this work is not properly worded and there many important works missing. Further, work needs a clear explanation. Besides, these two specific comments, the reviewer believes to major changes are required. 1) referencing is very incomplete and inaccurate at times and needs careful revision. Key papers on each topic should be added. 2) there should be a table or graphical representation showing the structure/organization of the table.  Detailed comments are listed below and need to be addressed carefully.   

The wording in the sentence “However, the existing literature is scarce in explicitly and systematically reviewing the application 11 of NOMA to different wireless networks. Therefore, the purpose of this survey is to fill this gap in ” Is not clear? What do you mean by “explicitly and systematically”? Aa a survey paper, the authors are expected to list existing survey/tutorial papers and discuss the difference of their work with them. While the authors have listed some relatively old survey papers such as [12], [14], [29], [26], they have missed newer and more relevant papers such as

[R1] M. Vaezi, et al., Interplay Between NOMA and Other Emerging Technologies: A Survey https://arxiv.org/pdf/1903.10489.pdf  

When talking about multiple access and NOMA, the authors may want to refer the readers to important contributions of the filed

[R2] Vaezi, M., Ding, Z., & Poor, H. V. Multiple access techniques for 5G wireless networks and beyond. Springer, 2019.

References need a clear review. Some references appear multiple times. For example. [11], [65], and [66] are basically one paper. This also suggests that each author has probably written one section and their works has not been combined properly. In sentence, “The existing survey papers on NOMA either present review from the perspective of different  NOMA classifications (such as power or code domain NOMA), or summarize state-of-the-art on integration of NOMA with different communication technologies (such as MIMO, cooperative systems, millimetre waves, and so on) [12,14,25–31]. However, to the best of our knowledge, there is no current  survey that systematically reviews the application of NOMA to different wireless systems. Therefore,  the goal of this survey is to exhaustively review the state-of-the-art on NOMA from the perspective  of its application to various types of wireless networks.” There are multiple issues: Again, what do you mean by “that systematically reviews the application of NOMA to different wireless systems” what is systematic in your paper that is not systematic in other surveys? Reference [30] and [31] are not survey papers at all and should not be cited here. Recent survey papers are missing Based on “In particular, Section II provides the basic concepts of NOMA. The application of NOMA to cellular, device-to-device (D2D), and wireless sensor networks have been comprehensively reviewed in Section III. The discussion of NOMA related research challenges and future directions is presented in Section IV. Finally, Section V concludes the survey.” The reviewer believes it is better the author clearly mentioned in the introduction that they are surveying the applications of NOMA in cellular, device-to-device (D2D), and wireless sensor networks. This paper is missing some of the very important papers in millimeter wave, massive MIMO, …. I understand those technologies have not been in the scope of this work but the abstract and introduction are written in a way that the survey is comprehensive. In multicell networks, the following pioneering works and many other references are missing:

[R3] W. Shin, et al., ‘‘Non-orthogonal multiple access in multi-cell networks: Theory, performance, and practical challenges,’’ IEEE Communications Magazine, vol. 55, no. 10, pp. 176–183, October 2017.

[R4] W. Shin, et al, ‘‘Coordinated beamforming for multi-cell MIMO-NOMA,’’ IEEE Communications Letters, vol. 21, no. 1, pp. 84–87, January 2017.

In future research, the authors need to better explain what they mean by “Alternate Receiver for NOMA”. In fact, SIC is motivated by the theory of the broadcast channel. That is, superposition coding with SIC achieves the capacity of the channel and if decoding changes we may not get the capacity region and NOMA will loose its spectral efficiency.

Round 2

Reviewer 1 Report

This survey had described the fundamental concept and potential benefits of NOMA protocol, which has been admitted as a new member of MA schemes. More precisely, the application of NOMA to CNs, D2D communications, and WSNs has been reviewed and discussed in detail. Moreover, outage probability expressions are presented and sum-rate maximization problems are formulated for each NOMA based wireless network. Furthermore, this study has also highlighted some of the key NOMA related design issues along with some potential future research directions.

Overall, the paper presents a reasonable scientific contribution. The suggested revisions have been properly incorporated and the paper is now well-structured.

Reviewer 2 Report

The reviewer's comments are mostly addressed.